# Dynamics of preventive vs post-diagnostic cancer control using low-impact measures

**Andrei R Akhmetzhanov[1,2], Michael E Hochberg[1,3,4]\***

[1]Institut des Sciences de l'Evolution de Montpellier, University of Montpellier, Montpellier, France; [2]Theoretical Biology Lab, Department of Biology, McMaster University, Hamilton, Canada; [3]Santa Fe Institute, Santa Fe, United States; [4]Wissenschaftskolleg zu Berlin, Berlin, Germany

**Abstract** Cancer poses danger because of its unregulated growth, development of resistance, and metastatic spread to vital organs. We currently lack quantitative theory for how preventive measures and post-diagnostic interventions are predicted to affect risks of a life threatening cancer. Here we evaluate how continuous measures, such as life style changes and traditional treatments, affect both neoplastic growth and the frequency of resistant clones. We then compare and contrast preventive and post-diagnostic interventions assuming that only a single lesion progresses to invasive carcinoma during the life of an individual, and resection either leaves residual cells or metastases are undetected. Whereas prevention generally results in more positive therapeutic outcomes than post-diagnostic interventions, this advantage is substantially lowered should prevention initially fail to arrest tumour growth. We discuss these results and other important mitigating factors that should be taken into consideration in a comparative understanding of preventive and post-diagnostic interventions.

**\*For correspondence:** mhochber@univ-montp2.fr

**Competing interests:** The authors declare that no competing interests exist.

## Introduction

Mathematical models play an important role in describing and analysing the complex process of carcinogenesis. Natural selection for increases in tumour cell population growth can be represented as the net effect of increased cell division rates and/or decreased apoptosis (e.g., *Wodarz and Komarova, 2007*). Relatively rare driver mutations confer such a net growth advantage, whereas numerically dominant passenger mutations with initially neutral or mildly deleterious effects (*Marusyk et al., 2012*; *Bozic et al., 2013*; *McFarland et al., 2013*) can increase in frequency due to genetic hitchhiking or subsequent positive selection. Amongst the many passengers in a growing tumour, some can contribute to chemoresistance, and sufficiently large tumours could contain different clones that, taken as a group, can resist some, if not most, chemotherapies (see *Michor et al., 2005* for resistance to imatinib). Chemotherapeutic remission followed by relapse suggests that these resistant cells are often present at low frequencies prior to therapy, either due to genetic drift or costs associated with resistance. Resistant phenotypes subsequently increase in frequency during radiotherapy or chemotherapy, and through competitive release they may incorporate one or more additional drivers, resulting in accelerated growth compared to the original tumour (for related discussion on pathogens, see *Huijben et al., 2013*).

Previous mathematical studies have considered alternatives to attempting to minimize or eradicate clinically diagnosed cancers with maximum tolerated doses (MTDs) of chemotherapeutic drugs. This body of work indicates that MTD is particularly prone to select for chemoresistance (e.g., *Foo and Michor, 2009*; *Foo and Michor, 2010*; *Lorz et al., 2013*), and what little empirical work exists supports this basic prediction (*Turke et al., 2010*), but see (*Kouyos et al., 2014*) for other disease systems. Numerous alternatives to the goal of cancer minimization/eradication have been proposed and investigated (e.g., *Maley et al., 2004*; *Komarova and Wodarz, 2005*; *Foo and Michor, 2009*;

**eLife digest** About one person in every two will get cancer during their lives. Surgery and chemotherapy have long been mainstays of cancer treatment. Both, however, have substantial downsides. Surgery may leave behind undetected cancer cells that can grow into new tumours. Furthermore, in response to chemotherapy drugs, some cancer cells may emerge that resist further treatment. There is therefore interest in whether preventive strategies—including lifestyle changes and medications—could reduce the likelihood of confronting a life-threatening cancer.

Now, Akhmetzhanov and Hochberg have developed a mathematical model to help compare the effectiveness of preventive strategies and traditional cancer treatments. The model—which assumes that a person can only develop a single cancer from a single region of pre-cancerous cells—suggests that long-term cancer prevention strategies reduce the risk of a life-threatening cancer by more than traditional treatment that begins after a tumour is discovered. The preventive measures may be less effective in some cases compared to traditional treatments if they initially fail to stop a tumour growing, although on average they still work better than treating the cancer after detection.

According to Akhmetzhanov and Hochberg's model, surgical removal followed by chemotherapy is less likely to be successful than prevention, and when successful, requires larger impacts on the cancer (and therefore creates more side-effects for the patient) to achieve the same level of control as prevention. The model also suggests that even at very low levels of impact on residual cancer cells, chemotherapies are likely to be counterproductive by boosting the subsequent emergence of treatment-resistant tumours.

Akhmetzhanov and Hochberg's model predicts how effective preventive measures need to be in terms of slowing the growth of cancer cells to result in given reductions in the future risk of a life-threatening cancer. Future work should test this model by measuring the effects on tumour growth of prevention and of traditional therapies.

*Gatenby et al., 2009a*, *2009b*; *Bozic et al., 2013*; *Jansen et al., 2015*). For example, *Komarova and Wodarz (2005)* considered how the use of one or multiple drugs could prevent the emergence or curb the growth of chemoresistance. They showed that the evolutionary rate and associated emergence of a diversity of chemoresistant lineages is a major determinant in the success or failure of multiple drugs vs a single one. Lorz and co-workers (*Lorz et al., 2013*) recently modelled the employment of cytotoxic and cytostatic therapies alone or in combination and showed how combination strategies could be designed to be superior in terms of tumour eradication or managing resistance than either agent used alone. *Foo and Michor (2009)* evaluated how different dosing schedules of a single drug could be used to slow the emergence of resistance given toxicity constraints. One of their main conclusions is that drugs slowing the generation of chemoresistant mutants and subsequent evolution are more likely to be successful than those only increasing cell death rates.

These and other computational approaches have yet to consider the use of preventive measures to reduce cancer-associated morbidity and mortality whilst limiting resistance. Prevention includes life-style changes and interventions or therapies in the absence of detectable invasive carcinoma (e.g., *Etzioni et al., 2003*; *Lippman and Lee, 2006*; *William et al., 2009*; *Hochberg et al., 2013*), for example, reduced cigarette consumption (*Doll and Peto, 1976*) or chemoprevention (*Steward and Brown, 2013*). In depth consideration of preventive measures and their likely impact on individual risk and epidemiological trends is important given the likelihood that all individuals harbour pre-cancerous lesions, some of which may transform into invasive carcinoma (*Bissell and Hines, 2011*; *Greaves, 2014*), and concerns as to whether technological advances will continue to make significant headway in treating clinically detected cancers (*Gillies et al., 2012*; *Vogelstein et al., 2013*).

Here, we model how continuous, constant measures affect tumour progression and the emergence of resistant lineages. We assume that an individual can contract at most a single cancer, originating from a single lesion. Importantly, we consider cases where the measure may select for the evolution of resistant phenotypes and cases where no resistance is possible. Our approach is to quantify the daily extent to which a growing neoplasm must be arrested in order to either eradicate it or to delay a potentially lethal cancer. Several authors have previously argued how constant or intermittent low toxicity therapies either before or after tumour discovery could be an alternative to MTD chemotherapies

(*Wu and Lippman, 2011*; *Hochberg et al., 2013*), but to our knowledge, no study has actually quantified based on empirical parameter estimates, the extent to which cancer cell population growth needs to be arrested for such approaches to succeed (see related discussion in *Bozic et al., 2010*; *Gerstung et al., 2011*; *Bozic et al., 2013*). Below we employ the terms 'treatment', 'measure', and 'therapy' interchangeably, all indicating intentional measures to arrest cancer cell population growth.

We first derive analytical expressions for the expected total number of cells within a tumour at any given time. We explore dynamics of tumour sizes at given times, and times to detection for given tumour sizes. Specifically, we show that the expected mean tumour size in a population of subjects can be substantially different from the median, since the former is highly influenced by outliers due to tumours of very large size. We then consider constant preventive measures and show that treatment outcome is sensitive to initial conditions, particularly for intermediate-sized tumours. Importantly, we provide approximate conditions for tumour control both analytically and numerically using empirical parameter estimates. We next consider post-diagnostic interventions in which tumour resection either is not complete and leaves residual cells or undetected metastases are present. We contrast these with prevention scenarios where (1) there is no difference in the age at which either prevention or post-diagnostic intervention commences, and (2) prevention and post-diagnostic interventions are alternatives, that is, the former always occurs before the latter. We show as expected that therapeutic outcomes are generally superior under prevention vs post-diagnostic intervention, and that higher impacts on the cancer cell population are usually required for post-diagnostic interventions to achieve a level of control comparable to prevention. Moreover, we find that should resection leave sufficiently large numbers of residual cells (or metastases are not discovered), then a range of the most successful outcomes under prevention is not attainable under post-diagnostic intervention, regardless of potential cell arrest. Finally and importantly, whereas there is little gained in terms of outcomes in post-diagnostic intervention beyond approximately 0.3% cell arrest per day for both small (10,000) and large (1 million) cancer cell populations, prevention outcomes may achieve continual gains for the latter cell number, up to about 0.6% cell arrest per day.

## Modeling framework

Previous study has evaluated the effects of deterministic and stochastic processes on tumour growth and the acquisition of chemoresistance (*Komarova and Wodarz, 2005*; *Bozic et al., 2010*; *Reiter et al., 2013*, see review *Beerenwinkel et al., 2015*). We first consider both processes through exact solutions and numerical simulations of master equations, using the mean field approach (see Appendix 1 for details). A mean field approach assumes a large initial number of cells (*Krapivsky et al., 2010*) and averages any effects of stochasticity, so that an intermediate state of the system is described by a set of ordinary differential equations (i.e., master equations; *Gardiner, 2004*). Solutions to these are complex even in the absence of the explicit consideration of both drivers and passengers (*Antal and Krapivsky, 2011*; *Kessler and Levine, 2013*).

We do not explicitly model the different pre-cancerous or invasive carcinoma states. Rather, our approach follows the dynamics of the relative frequencies of subclones, each composed of identical cells (*Baake and Wagner, 2001*; *Saakian and Hu, 2006*). We simulate tumour growth using a discrete time branching process for cell division (*Athreya and Ney, 1972*; *Bozic et al., 2010*). For each numerical experiment, we initiate a tumour of a given size and proportion of resistant cells.

Briefly, the model framework is as follows. Each cell in a population is described by two characteristics. The first is its resistance status to the measure, which is either 'not resistant' ($j = 0$) or 'resistant' ($j = 1$). The second property is the number of accumulated driver mutations (maximum $N$) in a given cell line. At each time step of 4 days, cells either divide or die, and when a cell divides, its daughter cell has a probability $u$ of producing a driver mutation and $v$ of producing a resistant mutation. We assume no back mutation, and that cells do not compete for space or limiting resources.

The fitness function $f_{ij}$, the difference between the birth and death rates of a cell, is defined by the number of accumulated drivers ($i = 0, 1, \ldots, N$) and resistance status ($j = 0, 1$): a sensitive cancerous cell with a single driver has selective advantage $s$, and any accumulated driver adds $s$ to fitness, while resistance is associated with a constant cost $c$. Exposure to a single treatment affects only non-resistant cells ($j = 0$), incurring a loss $\sigma$ to their fitness. Thus, the fitness function is:

$$f_{ij} = s(i+1) - \sigma(1-j) - cj.$$

The assumption of driver additivity is a special case of multiplicative fitness, and both are approximately equivalent for very small $s$.

We conducted numerical experiments, each with the same initial states but each using a unique set of randomly generated numbers of a branching process. For each simulation and each time step, the number of cells at time $(t+1)$ was sampled from a multinomial distribution of cells at time $t$ (see *Bozic et al., 2010* for details). *Table 1* presents baseline parameter values employed in this study. Hereafter, we refer to $\sigma$ as the treatment intensity (applied once every cell cycle of 4 days), while the corresponding daily arrest level to non-resistant cells is approximated by $\sigma/4$.

## Results

### Preventive measures

We first study preventive interventions where a patient has a high risk of developing a cancer and/or a biomarker that indicates the probable presence of a cancer. In either case, so that we can compare and contrast different intervention levels, we assume that the (undetected) tumour contains $M_0$ cells when prevention commences. We examine effects on the mean by considering the distribution of tumour sizes at different times using mean-field dynamics (see Appendix 1). Numerical experiments were then conducted by assuming that tumours initially contained $M_0 = 10^6$ identical cells ($i = 0$), of which 0.01% were resistant. These assumptions are obviously oversimplifications, and we relax some of them below and in the next sections.

There is an excellent correspondence between analytical and numerical results for $\sigma$ varied in range of $s$ (*Appendix 1—figure 1A*). A more detailed study of the distribution of tumour sizes reveals that the mean diverges considerably from median behaviour in the majority of cases, since the former is strongly influenced by outliers with high-tumour cell numbers (see *Appendix 1—figure 1B*).

*Figure 1* shows four examples of numerical experiments. An untreated tumour reaches the assumed detection threshold of $10^9$ cells by about 18 years on average and because it is not subject to strong negative selection (we assume low $c$), any emerging resistant cell-lines are likely to remain at low frequency (0.03% at the detection time in the example of *Figure 1A*). In *Figure 1B*, low-treatment intensity delays tumour growth and thus time of detection by approximately 16 years, while an increase in dose tends to result in tumours dominated by resistant cells (*Figure 1C*). Despite being unaffected by treatment, resistant cell populations are sometimes observed to go extinct stemming

**Table 1**. Baseline parameter values used in this study

| Parameter | Variable | Value | Range | Ref. |
|---|---|---|---|---|
| Time step (cell cycle length) | $T$ | 4 days | 3–4 days | (*Bozic et al., 2010*) |
| Selective advantage | $s$ | 0.4% | 0.1–1.0% | (*Bozic et al., 2010*) |
| Cost of resistance | $c$ | 0.1% | | |
| Mutation rate to acquire an additional driver | $u$ | $3.4 \times 10^{-5}$ | $10^{-7}$–$10^{-2}$ | (*Bozic et al., 2010*) |
| Mutation rate to acquire resistance | $v$ | $10^{-6}$ | $10^{-7}$–$10^{-2}$ | (*Komarova and Wodarz, 2005*) |
| Maximal number of additional drivers | $N$ | 5 (*Figures 1, 2*) 9 (other figures) | 0–9 | |
| Initial cell population | $M_0$ | $10^6$ cells | – | |
| Pre-resistance level | $\kappa$ | 0.01% | – | (*Iwasa et al., 2006*) |
| Number of replicate numerical simulations (excluding extinctions) | – | $10^6$ | – | |
| Detection threshold | $M$ | $10^9$ cells | $10^7$–$10^{11}$ | (*Beckman et al., 2012*) |

'Range' is values from previous study and employed in the present study.

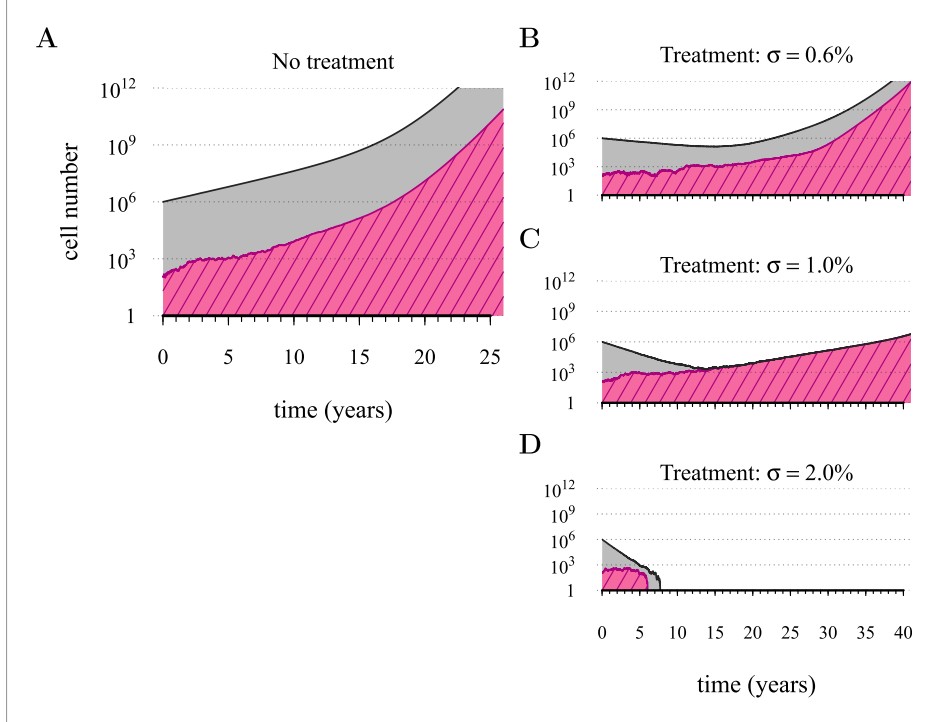

**Figure 1**. Treatments curb or eliminate tumours. Examples of single patient tumour growth for (**A**) no treatment. (**B**) $\sigma$ = 0.6%. (**C**) $\sigma$ = 1.0%. (**D**) $\sigma$ = 2.0%. The shaded area shows the change in total tumour size and the hatched area, the resistant part of a tumour. The treatment intensity $\sigma$ in this and all other figures are represented as cell arrest per day ($\sigma$/4). Parameter values as in **Table 1**.

from stochasticity (**Figure 1D**), and this tends to occur more at high-treatment levels, because there are fewer sensitive tumour cells to seed new (mutant) resistant cell populations.

We next considered how therapies affected the distribution of tumour detection times in cases where the cancer cell population attained a threshold of $10^9$ cells. The magnitude of the selective advantage $s$ shows that tumour growth is largely driven by its non-resistant part for relatively low-impact treatments $\sigma < 2s$ (**Figure 2A**). Importantly, the tumour shifts from being mainly non-resistant to resistant at $\sigma \approx 2s$, which is reflected by the inflection point in the trajectory of the median (indicated by point $B$ in **Figure 2A,B**). Notice that detection times are also most variable at $\sigma \approx 2s$. The median changes smoothly at high-treatment levels ($\sigma > 2s$), tending to a horizontal asymptote. This is explained by the fact that the sensitive part is heavily suppressed at high-treatment levels, meaning that the dynamics are strongly influenced by the actual time point at which the first resistance mutation occurs.

We find, counterintuitively, that early-detected tumours are more likely to be resistant under constant treatments than those detected at later times ($A$, $B$, and $C$ in **Figure 2C**). This is because tumours under treatment that by chance obtain resistance early grow faster than those that do not. By the time of detection, non-resistant tumours usually accumulate up to 4 additional drivers on average, while resistant tumours have fewer. For larger values of cost $c$, an additional non-regularity emerges at $\sigma \approx 3s$ (segment $DEF$ in **Figure 2B**), and is associated with tumours having a majority of cells with maximum numbers of drivers. This region is also characterized by a different transition to complete resistance (cf. **Videos 1**, **2** for relatively low and high costs of resistance, respectively). For example, at point $D$, tumours with a majority of non-resistance have less variable detection times than tumours with a majority of resistant cells (points $E$ and $F$ in **Figure 2B** and corresponding panels in **Figure 2C**). Treatment levels along the segment $DEF$ result in tumours that are more likely to be resistant as one goes from the centre to the tails of the distribution of detection times. This differs qualitatively from the previous case of a lower cost of resistance, where the tumours are less resistant in the tail of the distribution of detection times (cf segments $ABC$ and $DEF$ in **Figure 2B** and corresponding panels in **Figure 2C**).

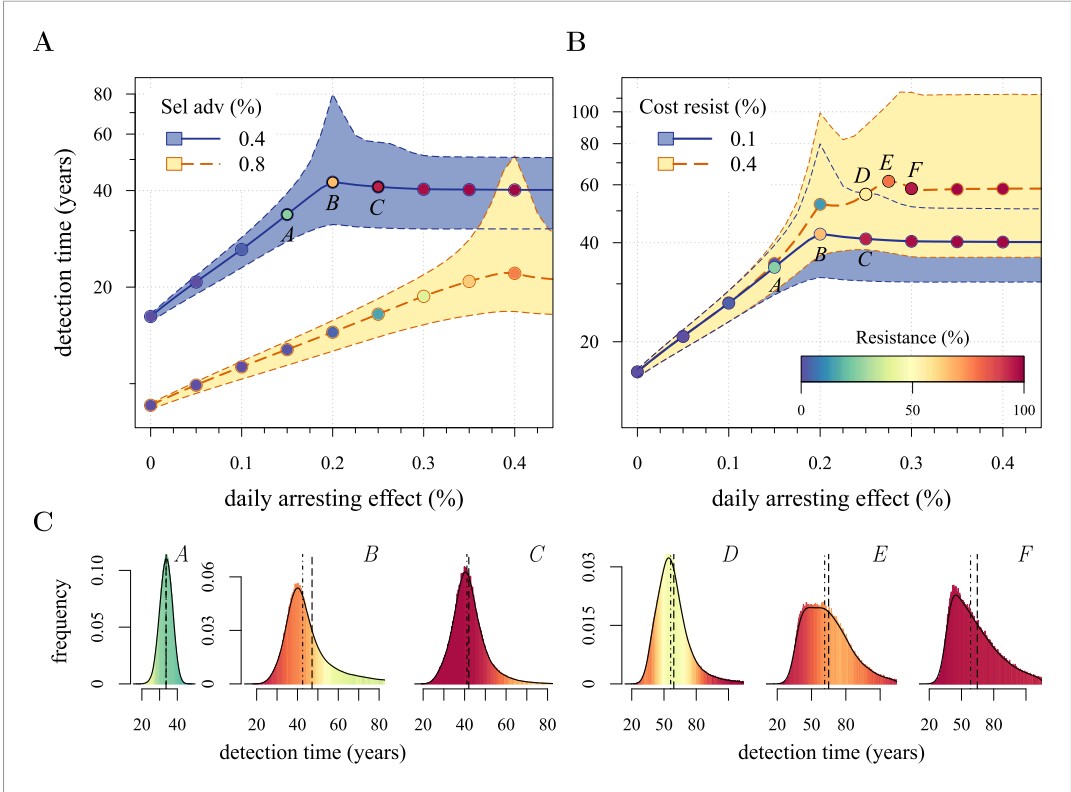

**Figure 2**. Treatment level affects both detection time and frequency of resistance. The median (lines) and 90% confidence intervals (shaded areas) of detection times, measured as years beyond the initiation of the preventive measure. Effects of: (**A**) the selective advantage of each additional driver and (**B**) the cost of resistance. (**C**) Samples of the distribution of detection times (in relative frequencies, adjusted for 3-month bins) for corresponding points, indicated in **A** and **B**. Dashed black line is the mean and the dashed-and-dotted line is the median. The colour-code indicates the average level of resistance in detected tumours over 3 month intervals (see inset in **B**). All cells $j = 0$ at $t = 0$. Other parameters as in **Table 1**. Detection time is log-transformed in **A** and **B**.

The following figure supplements are available for figure 2:

**Figure supplement 1**. Sensitivity analysis for several key parameters.

**Figure supplement 2**. Effects of initial neoplasm size (**A**, **B**) and resistance level (**C**) on preventive measure success.

The inflection point at $\sigma \approx 2s$ in **Figure 2A** is due to the accumulation of additional drivers within tumours and associated increases in the likelihood that the tumour eventually resists treatment. Since the initial population consists of $10^6$ cells, in the absence of treatment, a mutant cell with one additional driver and associated fitness $2s$ will appear very rapidly. Such a tumour can be suppressed only if $\sigma > 2s$. This is supported by additional numerical experiments where we vary the maximal number of additional driver mutations $N$: the inflection point $\sigma \approx 2s$ disappears when $N = 0$ (**Figure 2—figure supplement 1A**). The inflection points at $\sigma = 3s$, $4s$ emerge at treatment levels that effectively suppress sensitive subclones with the most drivers before resistance mutations are obtained (cf **Figure 2—figure supplement 1A–C** with **Figure 2—figure supplement 1D** and **Video 3**). Specifically, the peaked distributions, corresponding to better therapeutic outcomes, tend to disappear when resistant subclones emerge.

The initial cancer cell number $M_0$ affects both the median and distribution of detection times (**Figure 2—figure supplement 1B**). For large initial tumours, growth is deterministic and exponential. As the initial size is decreased from $10^6$ to $10^5$, stochastic effects are increasingly manifested by greater variability in tumour inhibition and an inflection point observed at the 95th percentile.

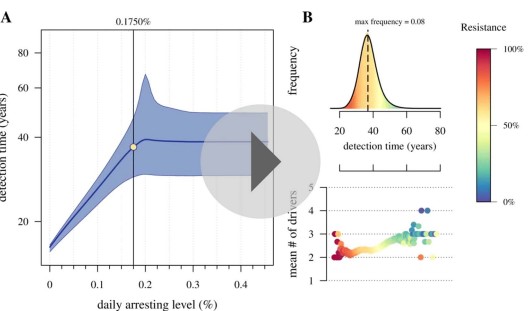

**Video 1.** Treatment level affects both detection time and frequency of resistance. (**A**) The median (thick line) and 90% confidence intervals (shaded areas with dashed boundaries) for the distribution of detection times. (**B**) Arbitrary samples of the distribution of detection times and distribution of the mean number of accumulated drivers. The colour-code indicates the average level of resistance in detected tumours over 3 month intervals. Parameters as in *Table 1*.

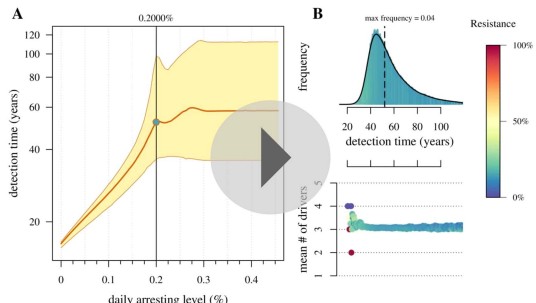

**Video 2.** Treatment level affects both detection time and frequency of resistance. (**A**) The median (thick line) and 90% confidence intervals (shaded areas with dashed boundaries) for the distribution of detection times. (**B**) Arbitrary samples of the distribution of detection times and distribution of the mean number of accumulated drivers. The colour-code indicates the average level of resistance in detected tumours over 3 month intervals. Parameters as in *Table 1* except for the cost of resistance $c = 0.4\%$.

Moreover, we find that a tumour is likely to be eradicated under a range of constant treatments when $M_0 = 10^5$ or fewer initial cells; in contrast, a tumour is virtually certain to persist regardless of treatment level for $M_0 = 10^7$ cells or greater (*Figure 2—figure supplement 2A,B*). In other words, our model indicates that tumours that are *c.* 1% the size of most clinically detectable, internal cancers will typically be impossible to eradicate by single molecule chemoprevention when resistance is possible.

Given the mutation rates assumed here, many tumours with 1 million cells will either already contain or rapidly subsequently acquire resistant cells (*Iwasa et al., 2006*). It is therefore not surprising that the initial fraction of resistant cells in a tumour has little impact on dynamics (*Figure 2—figure supplement 1C*). In contrast, another measure of success in control (the fraction of persons with tumours that remain undetected after 50 years of growth) improves substantially with lower numbers of initial resistance mutations, particularly at higher treatment levels (*Figure 2—figure supplement 2C*). This is because the initial phases of treatment have a major impact on the potential for new resistant mutants: should few be initially present or emerge, they will either go stochastically extinct or will not grow to detection levels (1 billion cells) in the 50 year time frame of these numerical experiments.

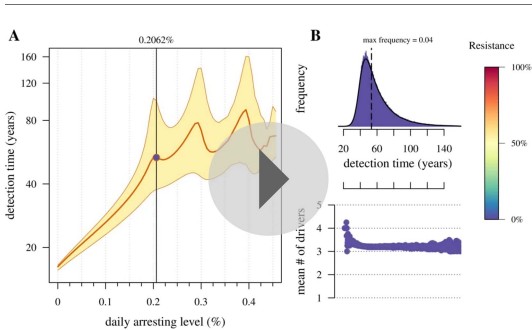

**Video 3.** Treatment level effects on detection times assuming no resistance is possible. (**A**) The median (thick line) and 90% confidence intervals (shaded areas with dashed boundaries) for the distribution of detection times. (**B**) Arbitrary samples of the distribution of detection times and the distribution of the mean number of accumulated drivers. The colour-code indicates the average level of resistance in detected tumours over 3 month intervals. The resistance mutation is knocked out ($v = 0$). Otherwise parameters as in *Table 1*.

We conducted further sensitivity analyses by varying accumulation rates $u$ of additional driver mutations. We find that tumours exhibit more or less deterministic growth depending on the initial number of cells $M_0$ and driver mutation rate $u$ whereby the larger the population (*Figure 2—figure supplement 1B*) or the higher the mutation rate (*Appendix 1—figure 4A*), the less apparent are stochastic effects. The corresponding analysis is presented in 'Varying mutation rate and initial tumour size' in Appendix 1 and *Appendix 1—figure 4*.

Finally, we considered scenarios where the cost of resistance is dose-dependent and specifically situations of drug addiction

(*Das Thakur et al., 2013*). Numerical studies presented in more detail in 'A simple form of drug addiction for resistant cell-lines' in Appendix 1 show that under dose-dependent costs, a drug treatment only applied when the number of non-resistant cells exceeds the number of resistant cells (e.g., a metronomic therapy [*Fischer et al., 2015*]) leads to slower long-term tumour growth than does a constant therapy.

## Post-diagnostic interventions

We next investigated how a post-diagnostic measure (usually some form of chemotherapy or radiation therapy, but could also involve adjuvants after an initial therapy) affects the probability of treatment success, the distribution of times for tumour relapse, and resistance levels. We assume that a tumour grows from one cell ($i = 0$, $j = 0$) and is discovered either at $10^9$ (early) or $10^{11}$ (very late) cells, whereupon the primary tumour is removed, leaving a small number ($10^4$ or $10^6$) of residual, and/or undetected or inoperable neighbouring micro-metastatic cells, and/or distant metastatic cells. Below, we contrast this with prevention without discriminating the age at which either intervention type commences, whereas in the following section, we consider these as competing alternatives. *Figure 3A* and *Figure 3—figure supplement 1A* present the distributions of driver mutations for each scenario. (Recall that in the previous section, we assumed that when a measure commenced, tumours had no additional drivers ($i = 0$)).

First, we examine the case where post-diagnostic resection leaves $10^6$ cells. As suggested by our studies above on prevention, 1 million cells have a high probability of already containing resistant subclones, and deterministic effects dominate subsequent tumour growth dynamics. Comparing the median expectations of years from tumour excision to relapse, early discovery (at $10^9$ cells) yields an additional 3.4 years compared to late discovery (at $10^{11}$ cells) at $\sigma = 1.5\%$ (medians for low vs high detection thresholds are 14.8 and 11.4 years, respectively; *Figure 3B*). Consider the following example: 20 years after resection and commencing treatment, the probability of tumour non-detection (i.e., the tumour is either eradicated or does not reach the detection threshold) is close to zero, regardless of treatment intensity (*Figure 3C*). Contrast this with cases of prevention starting at the same cancer cell population size ($10^6$ cells) but which fail to control the incipient tumour for the 50 years of the simulation: the detection time of these potentially life-threatening tumours is substantially longer than either of the excision cases (median 25.5 years for $\sigma = 1.5\%$, i.e., 0.3–0.4% potential cell arrest per day), and tumours are managed below the detection threshold after 20 years in more than 80% of cases for any $\sigma > 1.0\%$ (*Figure 3C*).

Now consider a residual population of 1/100th the previous case, that is, $10^4$ cells. Here, stochastic effects play a more important role in dynamics (*Figure 3—figure supplement 1A,B*). Due to initial heterogeneity (i.e., the co-occurrence of many subclones), when there are 4 and 5 (5 and 6) additional drivers in the dominant subclones of a residual cancer from an excised tumour of $10^9$ ($10^{11}$) cells, we observe a double peak at $4s$ and $5s$ ($5s$ and $6s$) (*cf Figure 3—figure supplement 1B*). These peaks in variability of outcomes are a result of the stochastic nature of the appearance of the first resistance mutations and of additional driver mutations. Interestingly, the secondary detection times (i.e., when residual or metastatic cells grow to form a new tumour) are more variable for small initial tumours compared to larger ones (*cf* the median 35.8 years, 90% CIs [17.0, 70.5] years vs 22.4, [13.7, 37.0] years for $10^9$ vs $10^{11}$, respectively, with $\sigma = 1.5\%$). This effect is due to resistance emergence in more aggressive subclones for larger tumours, such that the tumour relapses more deterministically (i.e., with less variability and faster on average). The probability of tumour non-detection after 20 years and the distribution of the mean number of accumulated drivers within tumours are shown in *Figure 3—figure supplement 1C,D*, respectively (*cf* with the previous case, shown in *Figure 3C,D*).

Importantly, for both thresholds of tumour excision, subsequent cancer cell arrest levels beyond approximately $\sigma = 1.5\%$ make little difference in terms of tumour growth (*Figure 3B-D*, *Figure 3—figure supplement 1B-D*), since virtually all of the sensitive cells post-excision will be arrested or killed by the measure beyond this level, leaving uncontrollable resistant cells to grow and repopulate the primary tumour site and/or metastases. (Note that this level is above that found in the previous section. This is because drivers accumulate throughout tumour growth in the results given in *Figure 3*, whereas tumours were assumed to only start accumulating the first drivers after growth from $M_0$ cells in *Figure 2* and *Figure 2—figure supplements 1, 2*). Moreover, we find that for post-diagnostic interventions knowledge about the number of drivers at the time of tumour discovery is a far better predictor of outcome than information about the time from tumour initiation to discovery, and that increases in treatment intensity tend to decrease predictive accuracy (*Figure 3—figure supplements 2–5*).

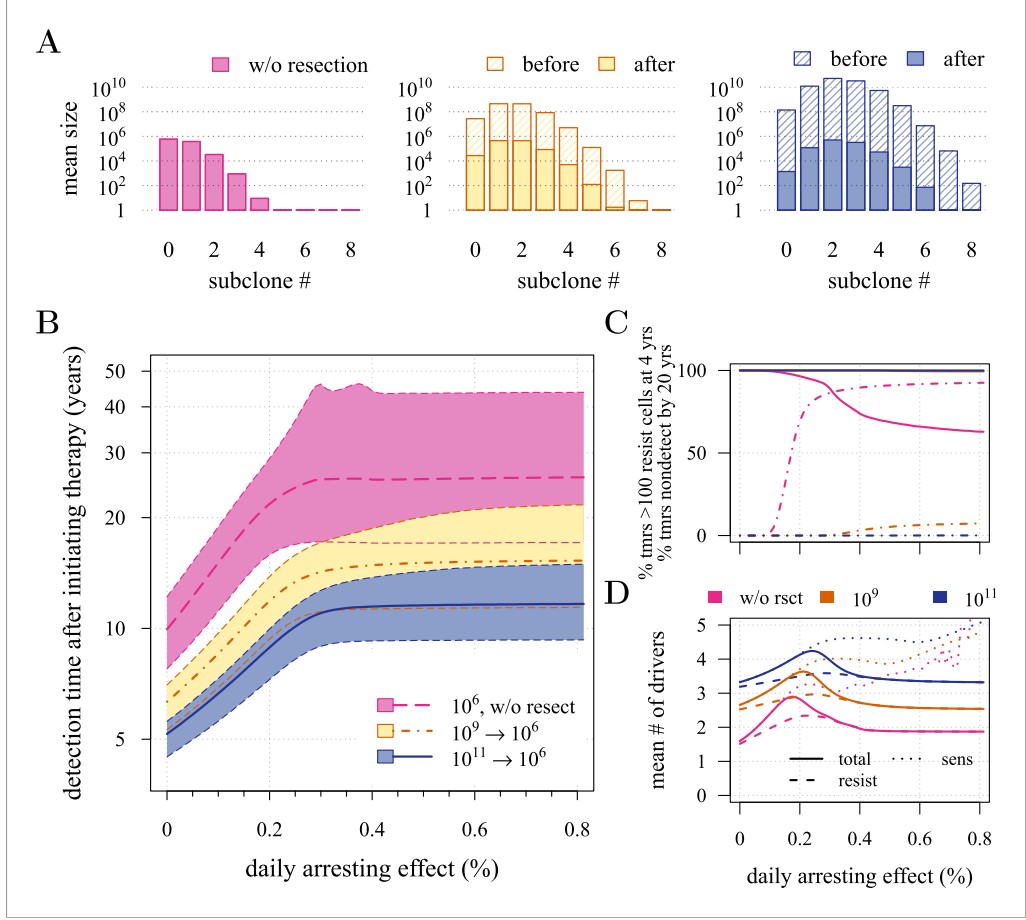

**Figure 3**. Effects of preventive and post-diagnostic interventions against tumours consisting of 1 million cells. (**A**) The distribution of mean sizes of subclones (hatched bars = before removal and solid bars = post removal). (**B**) The time distribution of cases in which either intervention type fails to control the tumour below the detection threshold after 50 years (thick line = median, filled area with dashed boundaries = 90% CIs) for different constant treatment intensities. (**C**) The percentage of cases where the tumour consists of less than 100 resistant cells at 4 years after treatment commences (solid lines), and the percentage of cases where tumour size is below the detection threshold 20 years after the measure begins (dashed-and-dotted lines). (**D**) The mean number of accumulated drivers within a tumour at the time of detection. Parameter values as in *Table 1*.

The following figure supplements are available for figure 3:

**Figure supplement 1**. Effects of preventive and post-diagnostic interventions against tumours consisting of 10,000 cells.

**Figure supplement 2**. Time to first discovery as a predictor of post-diagnostic treatment success.

**Figure supplement 3**. The R2 of regressions from numerical experiments for different treatment levels of time to tumour relapse following resection as function of the mean number of drivers in a resected tumour.

**Figure supplement 4**. Mean number of additionally accumulated drivers in resected tumour as a predictor of post-diagnostic treatment success.

**Figure supplement 5**. The $R^2$ of regressions from numerical experiments for different treatment levels of time to tumour relapse following resection as function of the mean number of drivers in a resected tumour.

## Prevention vs post-diagnostic intervention

The above results consider preventive measures and post-diagnostic interventions as independent rather than alternative approaches. Thus, although prevention delays tumour growth for longer times on average than does post-diagnostic intervention, because prevention is *always* initiated before diagnosis, when considering the relative benefits and risks of each, the actual time gained by the former relative to the latter in terms of cancer-free life will be less than the differences reported in *Figure 3B* and *Figure 3—figure supplement 1B*.

*Figure 4* presents a hypothetical comparative scenario of prevention vs post-diagnostic intervention. Prevention may either succeed without recurrence, or should the measure initially fail and a tumour be clinically detected, the patient has a 'second chance' whereby the tumour is resected and treatment

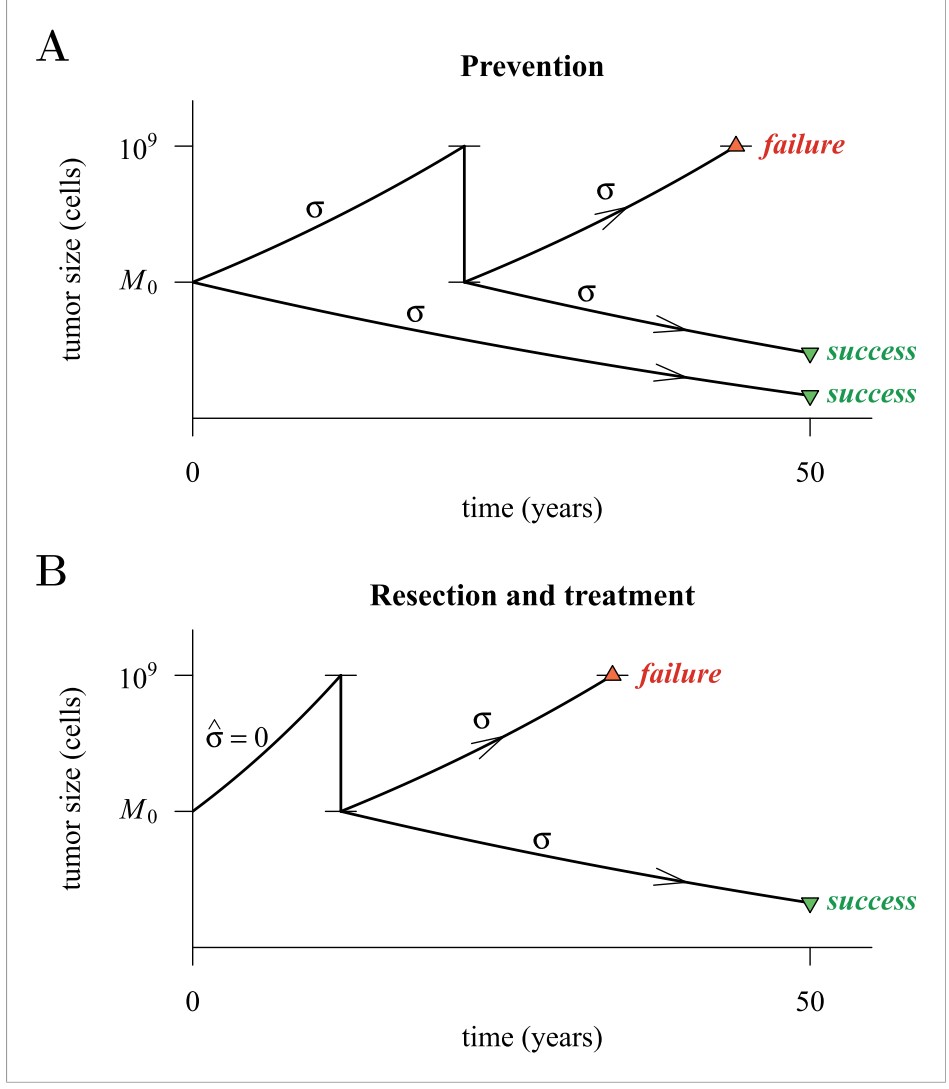

**Figure 4**. Hypothetical process of preventive (with a 'second chance') and post-diagnostic measures. A tumour is initiated by one cell and grows to size $M_0$ (either $10^4$ or $10^6$ cells in our numerical studies). Prevention (**A**) arrests tumour growth at intensity $\sigma$ (daily level = $\sigma/4$). Should the tumour grow to $10^9$ cells, it is diagnosed and resected to $M = M_0$ cells and then treated again at intensity $\sigma$. Post-diagnostic intervention (**B**) does not discover the growing tumour until $10^9$ cells (i.e., $\sigma = \hat{\sigma} = 0$), whereupon it is resected to $M = M_0$ cells and then treated at intensity $\sigma > 0$. Either intervention finally 'fails' should the tumour attain $10^9$ cells a second time, no later than 50 years after the initial lesion of size $M_0$. Should the tumour be eradicated or not exceed $10^9$ cells by 50 years after the initial lesion, then the intervention is deemed a 'success'.

continued (assumed at the same treatment intensity $\sigma$), either to a further relapse (failure) or non-detection (success) (*Figure 4A*). Compare this scenario with the more standard post-diagnostic resection followed by treatment, which either results in relapse or detection-free life (*Figure 4B*). These numerical experiments assume the same starting point (time at which the cancer cell population equals $M_0$, and drivers and resistant subclones are present) for each tumour, and because of a 'second chance' following initial failure in prevention, are run for a maximum of 50 years after the starting point (same as the numerical studies in the previous section). We also assume, as before, that potential therapeutic resistance mechanisms to all intervention types are identical.

*Figure 5* presents the comparative outcomes (see also *Videos 4*, *5*). When prevention starts at (or tumour resection misses) relatively large cancer cell populations (1 million cells), only small comparative gains occur from higher cell arrest in terms of outright treatment success (*Figure 5A*), whereas interventions starting at much smaller cancer cell numbers (10,000) result in considerably greater outright success (*Figure 5B*). Looking at situations of relapse only for prevention vs post-diagnostic intervention, the former generally results in superior outcomes in terms of delaying tumour growth, particularly for large residual cell populations (*cf Figure 5C,D*). In contrast, for lower numbers of residual cells, some post-diagnostic resected tumours in the sample will be initially resistance free (*cf Figure 5—figure supplement 1A,B*). This, together with fewer accumulated drivers in the highest driver subclones, contributes to improved outcomes should relapse occur (*Figure 5D*) and overall treatment success at sufficiently high treatment intensities (*Figure 5A,B,E,F*). Importantly, resected tumours in both the prevention (when it initially fails) and post-diagnostic scenarios may contain numerous resistant cells (example of 0.25% daily cellular arrest: *Figure 5—figure supplements 2, 3*). Prior selection for resistance in initially failed prevention generally results in larger residual resistant cell populations than pre-therapeutic residual populations in post-diagnostic situations (filled bars, *cf* captions A and B in *Figure 5—figure supplements 2, 3*), but smaller residual resistant cell populations than treatment failures following post-diagnostic resection (hatched bars, *cf* captions A and D in *Figure 5—figure supplements 2, 3*). Note that, as expected, secondary failures are associated with larger percentages of resistant subclones and a shift in the distributions towards more drivers (*cf* captions C and D in *Figure 5—figure supplements 2, 3*).

*Figure 5E,F* shows the distributions of detection times for all numerical experiments. We see that when both non-relapse (*Figure 5A*) and relapse (*Figure 5C*) are taken into account for large cancer cell populations (1 million cells), treating preventively at levels beyond about 0.3% arrest per day increases median delays in detection times due to outright success (i.e., survival beyond 50 years) but has no effect on the lower 95th percentile (*Figure 5E*). (Although not shown, arrest beyond approximately 0.6% per day does not yield further gains). In contrast, post-diagnostic intervention improves only marginally beyond daily arrest levels of about 0.3% (*Figure 5E*). *Figure 5F* shows the corresponding results for smaller cancer cell populations (based on integrating the results in *Figure 5B,D*), whereby a high median probability of full success is obtained >0.1% and >0.3% daily arrest for prevention and post-diagnostic intervention, respectively (*Figure 5F*). Thus for both cell population levels, prevention generally results in better outcomes compared to post-diagnostic intervention.

## Discussion

MTD chemotherapies present numerous challenges, a major one being the selection of resistant phenotypes, which are possible precursors for relapse (*Gerlinger and Swanton, 2010*). We mathematically and numerically investigated how the intensity of an anti-cancer measure, modelled as the arresting effect on a cancer cell population, resulted in success (i.e., either eradication or long-term tumour control) or failure (tumours growing beyond a threshold indicative of a life threatening cancer). Our central result is that beyond low impact thresholds—approximated by the Darwinian fitness of the subclone with the most driver mutations—little additional control is achieved when resistant subclones are present or likely to emerge during the long-term intervention assumed here.

We considered two contrasting scenarios. In the first, people at high risk of contracting a life threatening cancer make life-style changes or receive continuous, chemopreventive therapies, and in the second, more usual situation, a tumour is discovered and removed, and the patient treated with specific cytotoxic or cytostatic chemicals and/or with radiation. We found that, as expected, to achieve a given outcome, prevention requires smaller effects on cancer cell populations of a given size than do post-diagnostic interventions, the latter having smaller probabilities of complete cure and

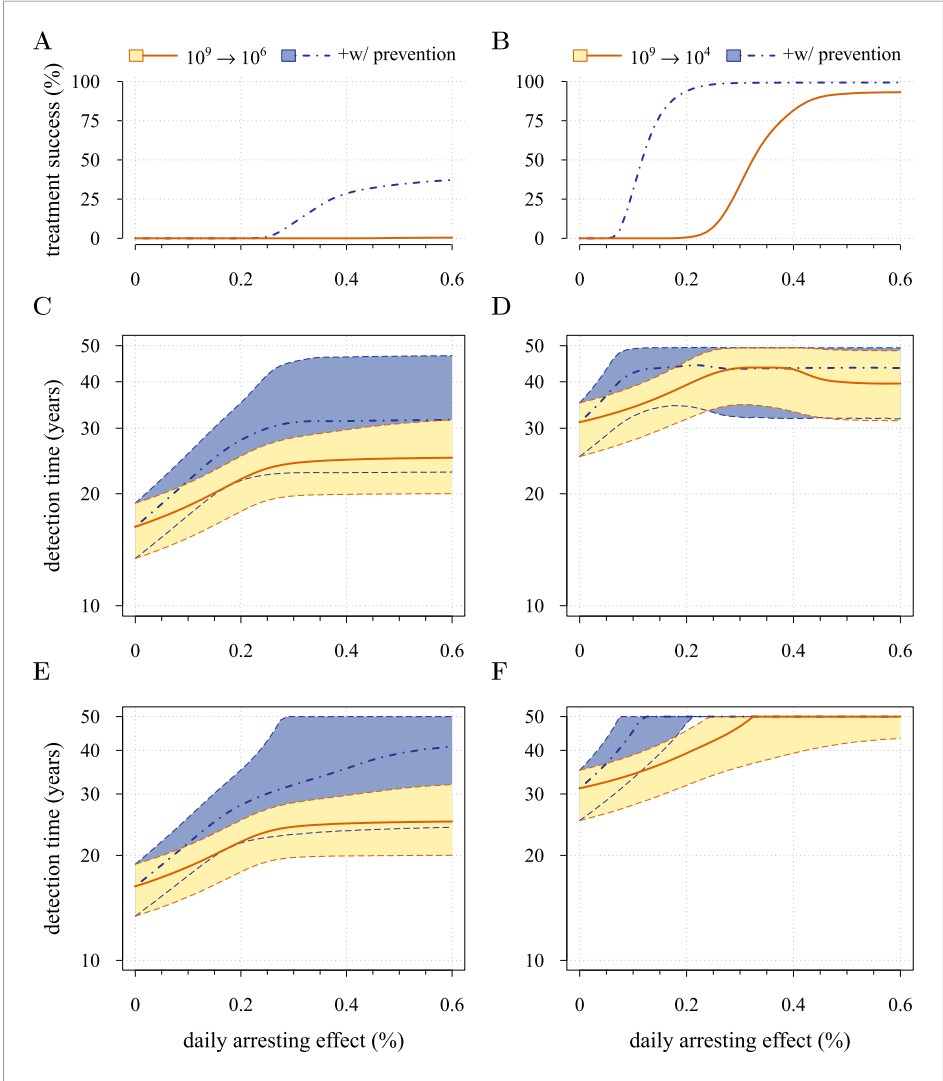

**Figure 5.** Comparison of preventive (blue lines and shading) and post-diagnostic (red lines, yellow shading) interventions. Tumours are either treated at $M_0 = 10^6$ cells (left panels) or $M_0 = 10^4$ cells (right panels). (**A**, **B**) Probability of treatment success, defined as the proportion of cases where the tumour remains undetected (either extinct or below $10^9$ cells) by 50 years after the initial lesion of $M_0$ cells. (**C**, **D**) Distribution of times to relapse for treatment failures. (**E**, **F**) Distribution of detection times for all cases including relapsed tumours and tumours remaining undetected prior to and after 50 years (detection times are assigned to 50 years in the latter case). Parameters as in *Table 1*. See *Figure 3* for details.

The following figure supplements are available for figure 5:

**Figure supplement 1**. Resistant cell populations after initial failure.

**Figure supplement 2**. Distribution of mean sizes of subclones.

**Figure supplement 3**. Distribution of mean sizes of subclones.

shorter times to tumour relapse. Inversely and importantly, for any given cell arrest level, prevention is, on average, superior to comparable post-diagnostic interventions, even when including cases where prevention initially fails, and resection and additional therapy are needed.

Specifically, based on empirical parameter estimates, we find that maximal long-term control occurs at surprisingly low daily levels of arrest. In the example where interventions target 1 million

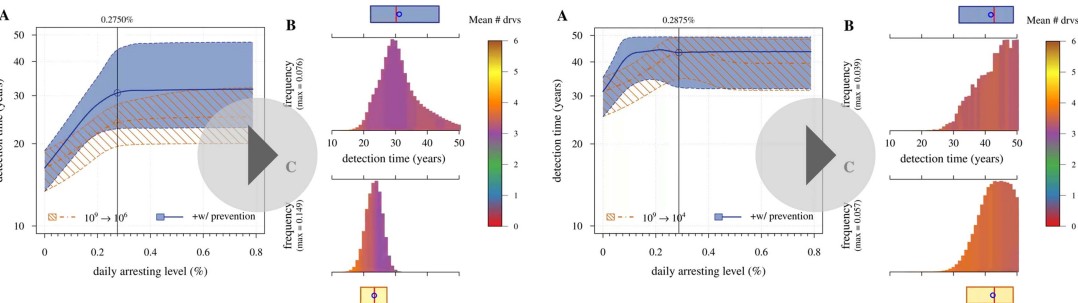

**Video 4.** Comparison of preventive (blue lines and shading) and post-diagnostic (red lines, hatched) interventions. Tumours are treated at $M_0 = 10^6$ cells. (**A**) The median (thick line) and 90% confidence intervals (shaded areas with dashed boundaries) for the distribution of times to relapse for treatment failures. (**B**) and (**C**) Arbitrary samples of the distribution of detection times for preventive and post-diagnostic interventions, respectively. The colour-code indicates the mean number of accumulated drivers over a period of 1 year. The rectangles on the top of **B** and on the bottom of **C** show the fifth and 95th percentiles, the blue circle indicates the median, and the red line is the mean. Parameters as in *Table 1*.

**Video 5.** Comparison of preventive (blue lines and shading) and post-diagnostic (red lines, hatched) interventions. Tumours are treated at $M_0 = 10^4$ cells. (**A**) The median (thick line) and 90% confidence intervals (shaded areas with dashed boundaries) for the distribution of times to relapse for treatment failures. (**B**) and (**C**) Arbitrary samples of the distribution of detection times for preventive and post-diagnostic interventions, respectively. The colour-code indicates the mean number of accumulated drivers over a period of 1 year. The rectangles at the top of **B** and the bottom of **C** shows the fifth and 95th percentiles, the blue circle indicates the median, and the red line is the mean. Parameters as in *Table 1*.

cancer cells, these levels are approximately 0.6% and 0.3% for preventive and post-diagnostic interventions, respectively. That the level is higher for preventive scenarios is because effective 'cure' (i.e., relapse does not occur during the 50 year period assumed in our numerical experiments) is possible, especially at cell arrest levels beyond 0.3%, whereas 'cure' is far less probable for post-diagnostic interventions. However, should prevention initially fail and a tumour be diagnosed and resected, any residual or metastatic cells are likely to contain more resistant clones than the corresponding situation for a post-diagnostic tumour. We stress that this latter result is contingent on our assumption that the same mutations (and mechanisms) are responsible for resistance to both preventive and post-diagnostic interventions. Should preventive and post-diagnostic measures differ substantially in their targets (and therefore resistance mechanisms), then evolved resistance to (failed) prevention could be irrelevant to the efficacy of subsequent traditional therapies.

Our results point to what is perhaps an underappreciated challenge in cancer control: low impact interventions risk being unable to control subclones with the most fitness-enhancing drivers, whereas high levels of arrest risk selecting for resistance (*Figure 6*). Future models should investigate these contingencies more extensively for alternative assumptions and a range of parameterizations for specific cancer types. Below, we discuss challenges to cancer management for both preventive and post-diagnostic scenarios.

## Preventive interventions

Whereas primary prevention is becoming an increasingly significant approach in reducing risk of certain cancers (e.g., *Colditz and Bohlke, 2014*), chemopreventive therapies are uncommon, despite empirical support for their effects (*William et al., 2009*). Several theoretical and in vitro experimental studies indicate that chemoprevention can reduce risks of life threatening cancers. For example, Silva and colleagues (*Silva et al., 2012*) parameterized computational models to show how low doses of verapamil and 2-deoxyglucose could be administered adaptively to promote longer tumour progression times. These drugs are thought to increase the costs of resistance and the competitive impacts of sensitive cells on resistant cancer cell subpopulations. However, some of the most promising results have come from studies employing non-steroidal anti-inflammatory drugs (NSAIDs), including experiments (*Ibrahim-Hashim et al., 2012*), investigations of their molecular effects (*Galipeau et al., 2007*; *Kostadinov et al., 2013*), and their use (*Cuzick et al., 2015*). For example,

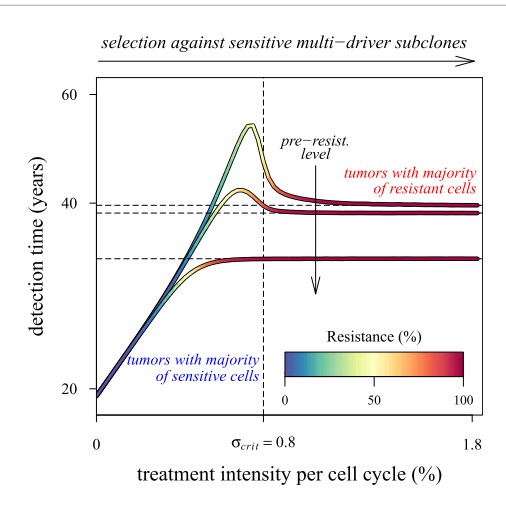

**Figure 6**. Dependence of the median time for tumour detection on treatment intensity and pre-resistance levels. Increasing treatment intensity selects against subclones with increasing numbers of drivers, whereas, regardless of treatment intensity, all resistant subclones with $s(i+1) > c$ increase in number. The solid lines illustrate how selection and the initial number of resistant cells in a treated tumour predict median detection times and associated resistance levels. Median detection times approach a horizontal asymptote at 100% resistance as treatment intensity increases, whereas if the resistant mutation were to be knocked out, then the vertical asymptote at $\sigma_{crit} = qs$ (where $q$ is the number of drivers in the fastest growing subclone) would be approached instead for sufficiently small tumours. Asymptotes are shown as dashed lines. We illustrate three cases, each with an initial population of 100,000 identical cells ($i = 0$) and with one of three different initial numbers of resistant cells: 10, 100 or 1,000 (top to bottom lines). Other parameters as in *Table 1*.

Ibrahim and co-workers (*Ibrahim-Hashim et al., 2012*) studied the action of NSAIDs and specifically sodium bicarbonate in reducing prostate tumours in male TRAMP mice (i.e., an animal model of transgenic adenocarcinoma of the mouse prostate). They showed that mice commencing the treatment at 4 weeks of age had significantly smaller tumour masses, and that more survived to the end of the experiment than either the controls or those mice commencing the treatment at an older age. *Kostadinov et al. (2013)* showed how NSAID use in a sample of people with Barrett's oesophagus is associated with reductions in somatic genomic abnormalities and their growth to detectable levels. It is noteworthy that it is not known to what extent reductions in cancer progression under NSAIDs are due to either cytotoxic or cytostatic effects or both. Although we do not explicitly model cytotoxic or cytostatic impacts, therapies curbing net growth rates but maintaining them at or above zero could be interpreted as resulting from the action of either cytotoxic and/or cytostatic processes. In contrast, therapies reducing net growth rates substantially below zero necessarily have a cytotoxic component. Our model, or modifications of it to explicitly include cytotoxic and cytostatic effects, could be used in future research to make predictions about optimal dose and start times to achieve acceptable levels of tumour control (or, e.g., the probability of a given tumour size and heterogeneity level by a given age).

Decisions whether or not to employ specific chemopreventive therapies carry with them the risk of a poorer outcome than would have been the case had another available strategy (or no treatment at all) been adopted (*Esserman et al., 2004*). This issue is relevant to situations where alterations in life-style, removal or treatment of pre-cancerous lesions, or medications potentially result in unwanted side effects or induce new invasive neoplasms (e.g., *Berrington de Gonzalez et al., 2011*). Chemopreventive management prior to clinical detection would be most appropriate for individuals with genetic predispositions, familial histories, elevated levels of specific biomarkers, or risk-associated behaviours or life-styles (*Hemminki and Li, 2004*; *Lippman and Lee, 2006*; *Sutcliffe et al., 2009*; *William et al., 2009*; *Hochberg et al., 2013*). Importantly, our approach presupposes that the danger a nascent, growing tumour presents is proportional to its size and (implicitly, all else being equal) a person's age. Due caution is necessary in interpreting our results, since studies have argued that metastatic potential rather than tumour size may be a better predictor of future survival (*Hynes, 2003*; *Foulkes et al., 2010*; *Sethi and Kang, 2011*). However, given the expectation that prevention typically confronts smaller, less heterogeneous neoplasms, which are less likely to have resistant clones and to have metastasised (*Hochberg et al., 2013*; *Gerlinger et al., 2014*), support our basic conclusion that prevention is generally a superior strategy in terms of cancer-free survival compared to post-diagnostic intervention.

## Post-diagnostic interventions

Over the past decade, several alternative approaches to MTD have been proposed, where the objective is to manage rather than eradicate tumours (e.g., *Maley et al., 2004*; *Komarova and Wodarz, 2005*;

*Gatenby, 2009*; *Gatenby et al., 2009a*, *2009b*; *Foo and Michor, 2010*; *Jansen et al., 2015*). Tumour management attempts to limit cancer growth, metastasis, and reduce the probability of obtaining resistance mutations through, for example, micro-environmental modification, or competition with non-resistant cancer cell populations or with healthy cells. These approaches usually involve clinically diagnosed cancers: either inoperable tumours or residual or metastatic cancers after tumour excision. In the former situation, tumours are typically large enough in size to contain numerous resistance mutations. In many, if not most, cases, these neoplasms will have metastasized, meaning greater variability both in terms of phenotypes and potential resistance to chemotherapies, and in penetrance of therapeutic molecules to targeted tumour cells (*Klein et al., 2002*; *Byrne et al., 2005*). In contrast, the latter situation involves smaller, residual, or metastatic cancer cell populations, composed of high frequencies of resistant variants or dormant cells (*Klein et al., 2002*). According to our results, both scenarios are likely to involve populations with large numbers of accumulated driver mutations (or, although not considered in our study, fewer driver mutations but each with larger selective effect), which ostensibly contribute to the speed of relapse. Thus, management of clinically detected tumours need not only limit the proliferation and spread of refractory subpopulations but should also aim to control the growth of multi-driver subclones (*Figure 5—figure supplements 2, 3*). In other words, in addition to actual resistance mutations ($j = 1$), subclones with $q$ drivers will be effectively resistant to therapeutic interventions if $q\,s \gg \sigma$ (*Figure 6*).

We therefore suggest that the frequency distribution of driver mutations and the distribution of resistant subclones within a heterogeneous cancer cell population could be used to instruct decisions of the time course of treatment levels, with the aims of curbing tumour growth, metastasis, and resistance. We found that tumours typically achieve several additional driver mutations by the time they reach detection (*Figure 3A*; *Figure 3—figure supplement 1A*; *Figure 5—figure supplements 2, 3*), which approximates certain estimates (*Stratton et al., 2009*) but falls short of others (*Sjoblom et al., 2006*).

## Conclusion

Our results indicate that the two most important variables in determining therapeutic outcome are (1) the size of the initial cancer cell population (i.e., when prevention commences and/or post-diagnosis, following resection), (2) associated tumour heterogeneity in terms of accumulated drivers, and the presence of resistance phenotypes. This highlights the importance of biomarkers as accurate indicators of otherwise undetectable malignancies (*Roukos et al., 2007*), and the accurate assessment of local or distant metastases (*Pantel et al., 1999*). We suggest that if order-of-magnitude estimates of cell populations and intra-tumour heterogeneity are possible, then low dose, continuous, constant approaches could be established that lower and possibly minimize risks of the emergence of future, life-threatening cancers. According to our model, such options will generally be superior to more aggressive chemotherapies if therapeutic resistance is a risk factor.

The framework proposed here is sufficiently general to portray major events in different types of cancer with emphasis on solid tumours. However, some aspects of cancerous tumour growth are considered only implicitly, and further research is required to formulate more realistic models to include, for example, spatial aspects of tumour growth (*Orlando et al., 2013*), competition/cooperation between different subclones (*Korolev et al., 2014*), combinational (multidrug) resistance (*Gillet and Gottesman, 2010*; *Bozic et al., 2013*), drug-addiction, observed for example in certain melanomas (*Das Thakur et al., 2013*), or advantageous resistant mutations, observed in some leukemias (*Michor et al., 2005*). Moreover, future studies should investigate alternatives to the traditional post-diagnostic therapeutic scenarios considered here (e.g., molecularly targeted therapies [*Yap, 2015*]). Our study nevertheless predicts that the main hurdle to post-diagnostic MTD interventions remains resistant subclones, since beyond minimal impacts on the order of 0.3% per day for the larger of the two residual or metastatic cell populations simulated here (which are still very small by clinical diagnostic standards—$c$ 1 mm³), increased therapeutic intensity selects disproportionally for resistance and has negligible benefits in terms of delaying life-threatening cancers.

## Acknowledgements

The authors are grateful to Athena Aktipis, Daniel Fisher, Sylvain Gandon, Urszula Hibner, Natalia Komarova, Patrice Lassus, Carlo Maley, Ville Mustonen, Robert Noble and anonymous referees for comments. All calculations were made using programs, written in *C*, and the free, open-source statistical

package *R* (*R Development Core Team, 2014*). The colour palette for figures was adopted from ColorBrewer 2.0 (*Brewer and Harrower, 2013*). Code for all calculations, and for producing all of the figures, is available at http://tiny.cc/AkhmHoch15Scripts and can be used freely for non-commercial purposes. The work was made possible by the facilities of the Shared Hierarchical Academic Research Computing Network (SHARCNET:www.sharcnet.ca) and Compute/Calcul Canada. ARA was supported by CNRS Interdisciplinary postdoctoral program. MEH thanks ITMO 'Physique Cancer' (CanEvolve PC201306), ANR (EvoCan ANR-13-BSV7-0003-01) and PICS (PlCS05313) for financial support.

## Additional information

### Funding

| Funder | Grant reference | Author |
|---|---|---|
| Institut national de la santé et de la recherche médicale | CanEvolve PC201306 | Michael E Hochberg |
| Agence Nationale de la Recherche | EvoCan ANR-13-BSV7-0003-001 | Michael E Hochberg |
| Centre National de la Recherche Scientifique | PICS 05313 | Michael E Hochberg |
| Réseau National des Systèmes Complexes | RNSC 2012 | Michael E Hochberg |

The funders had no role in study design, data collection and interpretation, or the decision to submit the work for publication.

### Author contributions

ARA, Performed computer simulations, Conception and design, Analysis and interpretation of data, Drafting or revising the article; MEH, Conception and design, Analysis and interpretation of data, Drafting or revising the article

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

## Appendix 1

### Mean-field approach

We use the mean-field approach (see for example, *Krapivsky et al., 2010*), which approximates the behaviour of a system consisting of many cells, so that the effects of stochasticity are averaged, and an intermediate state is described by a set of ordinary differential equations.

### Master equations

We write master equations to track the probability $P_{ij}(t)$ that a randomly chosen cell from a population of tumour cells is of type $(i, j)$ at time $t$.

The temporal dynamics of probabilities $P_{ij}(t)$, $i = 0, 1, \ldots, N$, where $N$ is the maximal number of additionally acquired drivers and $j = 0, 1$, are described by:

$$\frac{dP_{ij}(t)}{dt} = \mathbb{P}_{ij} + u\mathbb{P}_{ij}^{(u)} + v\mathbb{P}_{ij}^{(v)}.$$

Here, the right-hand side is a superposition of probabilistic in- and out-flows from different mutational states to the current one $(i, j)$. The function $\mathbb{P}_{ij}$ describes the growth of subclone $(i, j)$ and is proportional to the probability $P_{ij}(t)$, multiplied by the difference between fitness $f_{ij}$ and its average value over the whole population $\bar{f}(t) = \sum_{i,j} f_{ij} P_{ij}(t)$. Functions $\mathbb{P}_{ij}^{(u)}$ and $\mathbb{P}_{ij}^{(v)}$ represent the probabilistic flows of mutations. For $\mathbb{P}_{ij}^{(u)}$, a driver is added from class $(i - 1, j)$ to $(i, j)$ in proportion to the probability $P_{i-1,j}(t)$, the probability of cell birth $b_{i-1,j}$, and the probability of a zero locus being chosen from $N$ total loci consisting of $(N - (i - 1))$ other zero loci. A similar approach is used to define the outflow term for the probability from class $(i, j)$ to $(i + 1, j)$. The second term $\mathbb{P}_{ij}^{(v)}$ is the probability of mutating to therapeutic resistance $(i, j = 0)$ to $(i, j = 1)$ and is proportional to $P_{i0}(t)$ and birth rate $b_{i0}$. Finally, all terms are summed, taking into account the initial conditions: $P_{00}(0) = 1 - \kappa$, $P_{01}(0) = \kappa$, and $P_{ij} = 0$ for any other $i$ or $j$.

The above elements lead to the following system of ordinary differential equations (ODEs):

$$\frac{dP_{ij}(t)}{dt} = \left(f_{ij} - \bar{f}(t)\right) P_{ij}(t) + u\left[\left(1 - \frac{i-1}{N}\right)\frac{1 + f_{i-1,j}}{2} P_{i-1,j}(t) - \left(1 - \frac{i}{N}\right)\frac{1 + f_{ij}}{2} P_{ij}(t)\right]$$
$$- v(1 - 2j)\frac{1 + f_{i0}}{2} P_{i0}(t), \tag{1}$$

where some probabilities $P_{ij}(t)$ could, theoretically, take on negative values, for example, $P_{-1,j}(t)$, when $i = 0$, in which case, they are set to zero.

A simple transformation,

$$p_{ij}(0) = P_{ij}(0), \quad p_{ij}(t) = P_{ij}(t)\exp\left(\int_0^t \bar{f}(r)\,dr\right),$$

allows omitting the term $\bar{f}(t)$ from *Equation 1* and to linearize the latter with respect to the new 'transformed' probabilities $p_{ij}(t)$. This gives:

$$\frac{dp_{ij}(t)}{dt} = f_{ij}p_{ij}(t) + u\left[\left(1 - \frac{i-1}{N}\right)\frac{1 + f_{i-1,j}}{2} p_{i-1,j}(t) - \left(1 - \frac{i}{N}\right)\frac{1 + f_{ij}}{2} p_{ij}(t)\right]$$
$$+ v\frac{1 + f_{i0}}{2}\left(jp_{i,j-1}(t) + (j - 1)p_{ij}(t)\right), \tag{2}$$

where, for convenience, we write $(jp_{i,j-1}(t) + (j - 1)p_{ij}(t))$ instead of $(1 - 2j)p_{i0}(t)$.

## Probability generating function approach

With **Equation 2** we apply the probability generating function (p.g.f.) method (**Gardiner, 2004**; **Assaf, 2010**) to transform the system of $(2N + 1)$ ODEs to a Hamilton–Jacobi (HJ) equation, that is, a first order partial differential equation.

We define the p.g.f. as the polynomial over all modified probabilities $p_{ij}(t)$ of the form:

$$G(\xi, \eta, t) = \sum_{i=0}^{N} \sum_{j=0}^{1} \xi^i \eta^j p_{ij}(t), \tag{3}$$

where $\xi$ and $\eta$ are the variables that can be viewed as the momentum of an auxiliary Hamiltonian system governing the leading-order stochastic dynamics of the system (**Elgart and Kamenev, 2004**). Notice that the function $G(\xi, \eta, t)$ is linear with respect to $\eta$.

Suppose that the function $G(\xi, \eta, t)$ is defined. One can then obtain all characteristics of the stochastic process, such as the average tumour size $n(t)$ and the average frequency $n_{res}(t)/n(t)$ of resistant cells within a tumour. The former quantity is:

$$\frac{dn(t)}{dt} = n(t)\bar{f}(t).$$

Using the normalization condition for the probability: $\sum_{i,j} P_{ij}(t) = 1$, we obtain:

$$G(\xi = 1, \eta = 1, t) = \exp\left( \int_0^t \bar{f}(r)dr \right),$$

and then:

$$n(t) = M_0 \exp\left( \int_0^t \bar{f}(r)dr \right) = M_0 G(\xi = 1, \eta = 1, t), \tag{4}$$

where the initial tumour size $n(0) = M_0$ is sufficiently large. The frequency of resistant cells is defined as follows:

$$\frac{n_{res}(t)}{n(t)} = \sum_{i=0}^{N} P_{i1}(t) = \sum_{i=0}^{N} p_{i1}(t)\exp\left( -\int_0^t \bar{f}(r)dr \right) = \left. \frac{\partial G/\partial \eta}{G(\xi, \eta, t)} \right|_{\substack{\xi=1, \\ \eta=1}}. \tag{5}$$

Initial conditions yield $p_{00}(0) = 1 - \kappa$, $p_{01}(0) = \kappa$, and $p_{ij}(0) = 0$ for any other $i$ and $j$, so that $G(\xi, \eta, t = 0) = 1 - \kappa + \kappa\eta$.

To obtain the HJ equation related to the p.g.f. $G(\xi, \eta, t)$, we multiply **Equation 2** with $\xi^i \eta^j$ and sum up all equations for $i = 0, 1,\ldots,N$ and $j = 0, 1$. After some algebra, we obtain:

$$\frac{\partial G}{\partial t} = \left[ s\left( \xi\frac{\partial}{\partial \xi} + 1 \right) - \sigma\left( 1 - \eta\frac{\partial}{\partial \eta} \right) - c\eta\frac{\partial}{\partial \eta} + \frac{u(\xi - 1)}{2}\left( 1 - \frac{\xi}{N}\frac{\partial}{\partial \xi} \right) + \frac{v(\eta - 1)}{2}\left( 1 - \eta\frac{\partial}{\partial \eta} \right) \right] G, \tag{6}$$

where only terms of order greater than or equal to $u$, $v$ are retained, meaning that terms composed of the products $s$, $c$, and $u$, $v$ are omitted.

**Equation 6** is solved by the method of characteristics such that the HJ equation is transformed into a system of ordinary differential equations (i.e., the system of characteristics, see e.g., **Melikyan, 1998**).

## Time-varied treatment schedule

We find the characteristics for the variables $\xi$ and $\eta$ using (**Equation 6**):

$$\frac{d\xi(t)}{dt} = -s\xi(t) + \frac{u\xi(t)(\xi(t)-1)}{2N}, \quad \frac{d\eta(t)}{dt} = (c-\sigma(t))\eta(t) + \frac{v\eta(t)(\eta(t)-1)}{2}, \tag{7}$$

where $\sigma(t)$ is a given function of time.

The p.g.f. $G(\xi, \eta, t)$ changes along the characteristic **Equation 7** according to the following ODE:

$$\frac{dG(t)}{dt} = \left(s - \sigma + \frac{u(\xi(t)-1)}{2} + \frac{v(\eta(t)-1)}{2}\right)G(t), \tag{8}$$

which is straightforward to integrate. Indeed, if we use **Equation 7**, this yields $d\ln G = (s(N+1) - c)dt + Nd\ln\xi + d\ln\eta$. Thus,

$$G(\xi, \eta, t)\exp[-(s(N+1)-c)t - N\ln\xi - \ln\eta] = const. \tag{9}$$

Recall that the quantity on the left hand side remains constant only along the characteristic curve (**Equation 7**).

To obtain $G(\xi = 1, \eta = 1, t)$, we need to solve **Equation 7** subject to $\xi(t) = \eta(t) = 1$ and find $\xi(0)$ and $\eta(0)$. Then, given the initial condition $G(\xi(0),\eta(0),0) = 1 - \kappa + \kappa\eta(0)$, $\kappa$ is a level of resistance within a tumour ($\kappa \in [0,1]$), we can define $G(\xi, \eta, t)$ using **Equation 9**.

Finally, we use **Equation 4** to derive the dynamics of $n(t)$. To obtain the mean frequency of resistant cells within a tumour, we first write $\partial G/\partial\eta$, using **Equation 9** with the right hand side implicitly dependent on $\eta$ and then substitute it into **Equation 5**. (Note that time $t$ is measured in cell cycles, which are assumed to be of 4 days on average. To derive all necessary equations with respect to the actual time, we need to divide $t$ by the length of the cell-cycle and substitute it in the equations.)

## Constant treatment

We study the case for constant $\sigma$. Notice that this includes the case of no treatment ($\sigma = 0$).

First, we find the characteristics for the variables $\xi$ and $\eta$. Namely, the solution of **Equation 7** gives:

$$\xi(0) = \frac{s + u/(2N)}{\left(\frac{s+u/(2N)}{\xi(t)} - \frac{u}{2N}\right)e^{-(s+u/(2N))t} + \frac{u}{2N}}, \quad \eta(0) = \frac{\sigma - c + v/2}{\left(\frac{\sigma-c+v/2}{\eta(t)} - \frac{v}{2}\right)e^{-(\sigma-c+v/2)t} + \frac{v}{2}}. \tag{10}$$

The subsequent substitution of **Equation 10** into **Equation 8** leads to:

$$G(\xi, \eta, t) = G(\xi(0), \eta(0), 0)\exp\left[\left(s - \sigma - \frac{u+v}{2}\right)t + N\ln\left(1 + \frac{\xi u}{2N}\frac{e^{(s+u/(2N))t}-1}{s+u/(2N)}\right)\right.$$
$$\left. + \ln\left(1 + \frac{\eta v}{2}\frac{e^{(\sigma-c+v/2)t}-1}{\sigma-c+v/2}\right)\right].$$

Taking into account $u, v \ll s, c$ and assuming $v \ll \sigma - c$, we simplify further and write its approximate form:

$$G(\xi, \eta, t) \cong \left(1 - \kappa + \frac{\kappa\eta e^{(\sigma-c)t}}{1 + \frac{\eta v}{2}\frac{e^{(\sigma-c)t}-1}{\sigma-c}}\right)\exp\left[(s-\sigma)t + N\ln\left(1 + \frac{\xi u}{2N}\frac{e^{st}-1}{s}\right) + \ln\left(1 + \frac{\eta v}{2}\frac{e^{(\sigma-c)t}-1}{\sigma-c}\right)\right],$$

which can be also written in the form:

$$G(\xi, \eta, t) \cong \left((1-\kappa)\left(1 + \frac{\eta v}{2}\frac{e^{(\sigma-c)t}-1}{\sigma-c}\right) + \kappa\eta e^{(\sigma-c)t}\right)\exp\left[(s-\sigma)t + N\ln\left(1 + \frac{\xi u}{2N}\frac{e^{st}-1}{s}\right)\right]. \tag{11}$$

As expected **Equation 11** is linear with respect to $\eta$.

Thus, we derive an analytical expression for the dynamics $n(t)$. Namely, we use **Equations 4** and **11** and substitute $\xi = \eta = 1$, to obtain:

$$n(t) = M_0 \left( (1-\kappa)\left(1 + \frac{v}{2}\frac{e^{(\sigma-c)t}-1}{\sigma-c}\right) + \kappa e^{(\sigma-c)t} \right) \exp\left[ (s-\sigma)t + N\ln\left(1 + \frac{u}{2N}\frac{e^{st}-1}{s}\right) \right]. \quad (12)$$

**Equation 12** is simplified for two limiting cases. In the early stages of tumour growth, the value $n(t)$ changes according to a hyper-exponential law:

$$n(t) \cong M_0 \left( (1-\kappa)\left(1 + \frac{v}{2}\frac{e^{(\sigma-c)t}-1}{\sigma-c}\right) + \kappa e^{(\sigma-c)t} \right) \exp\left( (s-\sigma)t + \frac{u}{2}\frac{e^{st}-1}{s} \right).$$

while at later stages, the most aggressive subclone persists, being sensitive if $\sigma < c$ ($n(t) \propto e^{s(N+1)t}$) and resistant otherwise ($n(t) \propto e^{(s(N+1)-c)t}$).

To compute the frequency of resistant cells within a tumour (**Equation 5**), we derive $\partial G/\partial \eta$ using **Equation 11**:

$$\frac{\partial G}{\partial \eta} = \left( (1-\kappa)\frac{v}{2}\frac{e^{(\sigma-c)t}-1}{\sigma-c} + \kappa e^{(\sigma-c)t} \right) \exp\left[ (s-\sigma)t + N\ln\left(1 + \frac{\xi u}{2N}\frac{e^{st}-1}{s}\right) \right].$$

so that:

$$\frac{n_{res}(t)}{n(t)} = \frac{(1-\kappa)\frac{v}{2}\frac{e^{(\sigma-c)t}-1}{\sigma-c} + \kappa e^{(\sigma-c)t}}{(1-\kappa)\left(1 + \frac{v}{2}\frac{e^{(\sigma-c)t}-1}{\sigma-c}\right) + \kappa e^{(\sigma-c)t}}.$$

*Appendix 1—figure 1* shows the excellent correspondence between numerical experiments and analytical results for $\sigma$ on the order of $s$. *Appendix 1—figure 2* provides additional details on distribution of tumour sizes depending on the applied treatment intensity (points $B$ and $C$).

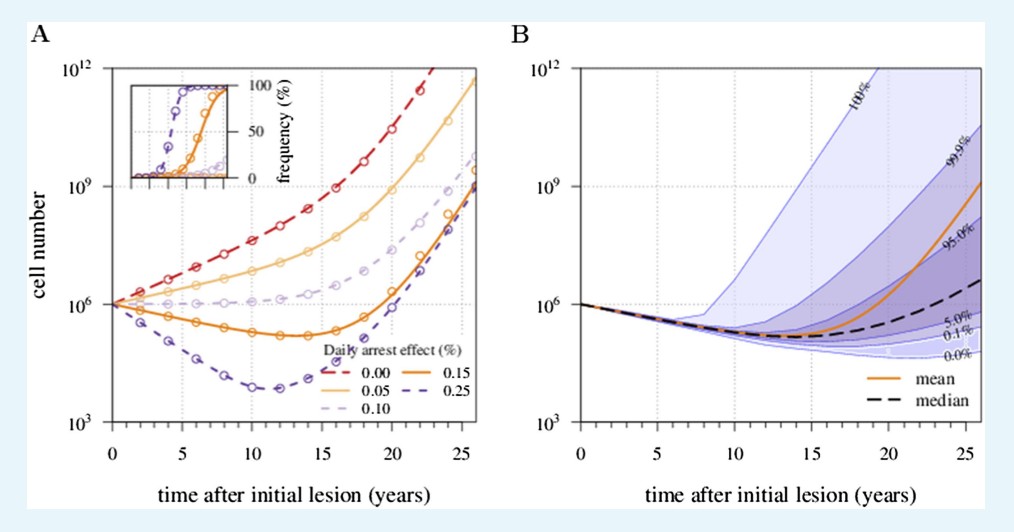

**Appendix 1—figure 1**. Mean field dynamics concord with numerical simulations. (**A**) Effect of treatment level and observation time on mean tumour size. (**Inset**) Mean frequency of resistant cells within tumours corresponding to three of the cases in **A**. Lines are analytically computed mean-field trajectories, while dots are numerical simulations (see Appendix 1 for details). (**B**) Dynamics of mean and median tumour size, and percentiles around the median (shaded areas), assuming a fixed constant arresting effect of 0.15 / day. Treatments start at $t = 0$, and the maximal number of additionally accumulated drivers is 3. See *Table 1* for other parameter values.

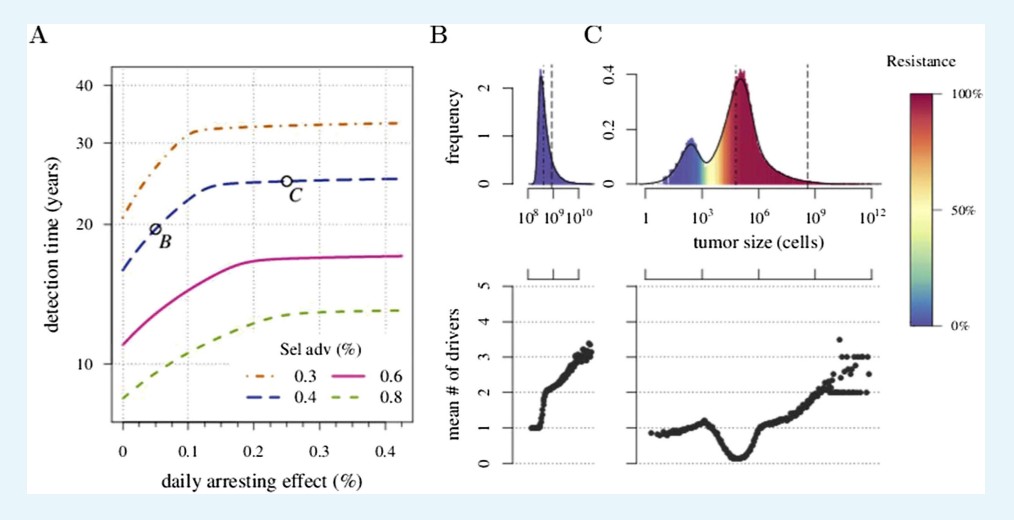

**Appendix 1—figure 2**. Trade-off between growth and resistance under different treatment regimes. (**A**) Analytically derived times for a tumour to reach $10^9$ cells (see *Equation 12*). (**B**) and (**C**) Sample distributions in relative frequencies, adjusted for bins over periods of 0.5 in logarithmic scale for corresponding points $B$ and $C$, shown in plot **A**. Dashed black line is the mean and the dashed-and-dotted line is the median. The bottom panel shows the mean number of additionally accumulated drivers for all detected tumours over the same intervals of 3 months. The colour-code indicates the level of resistance in detected tumours over these intervals. Maximal number of additionally accumulated drivers is 5. Parameters otherwise as in *Table 1*.

## Distribution of subclones within an exponentially growing tumour

The p.g.f. $G(\xi, \eta, t)$ is used to derive expressions for all $P_{ij}(t)$, which are the probabilities of selecting a cell of type $(i, j)$ from a tumour at time moment $t$. Namely, we need to differentiate the p.g.f. with respect to $\xi$ and $\eta$, so that:

$$P_{ij}(t) = \frac{1}{i!\, G(\xi=1, \eta=1, t)} \frac{\partial^{i+j} G(\xi=0, \eta=0, t)}{\partial \xi^i \partial \eta^j},$$

where $i = 0, 1, \ldots$ and $j = 0, 1$. Thus, we write:

$$P_{00}(t) = \frac{G(0,0,t)}{G(1,1,t)} = \frac{1-\kappa}{(1-\kappa)\left(1 + \frac{v}{2} \frac{e^{(\sigma-c)t} - 1}{\sigma - c}\right) + \kappa e^{(\sigma-c)t}} \left(1 + \frac{u}{2N} \frac{e^{st} - 1}{s}\right)^{-N},$$

$$P_{01}(t) = \frac{\partial G(0,0,t)/\partial \eta}{G(1,1,t)} = \frac{(1-\kappa)\frac{v}{2} \frac{e^{(\sigma-c)t} - 1}{\sigma - c} + \kappa e^{(\sigma-c)t}}{(1-\kappa)\left(1 + \frac{v}{2} \frac{e^{(\sigma-c)t} - 1}{\sigma - c}\right) + \kappa e^{(\sigma-c)t}} \left(1 + \frac{u}{2N} \frac{e^{st} - 1}{s}\right)^{-N},$$

then:

$$P_{10}(t) = \frac{\partial G(0,0,t)/\partial \xi}{G(1,1,t)} = \frac{1-\kappa}{(1-\kappa)\left(1 + \frac{v}{2} \frac{e^{(\sigma-c)t} - 1}{\sigma - c}\right) + \kappa e^{(\sigma-c)t}} \frac{u}{2} \frac{e^{st} - 1}{s} \left(1 + \frac{u}{2N} \frac{e^{st} - 1}{s}\right)^{-N},$$

$$P_{11}(t) = \frac{1}{G(1,1,t)} \frac{\partial^2 G(0,0,t)}{\partial \xi \partial \eta} = \frac{(1-\kappa)\frac{v}{2} \frac{e^{(\sigma-c)t} - 1}{\sigma - c} + \kappa e^{(\sigma-c)t}}{(1-\kappa)\left(1 + \frac{v}{2} \frac{e^{(\sigma-c)t} - 1}{\sigma - c}\right) + \kappa e^{(\sigma-c)t}} \frac{u}{2} \frac{e^{st} - 1}{s} \left(1 + \frac{u}{2N} \frac{e^{st} - 1}{s}\right)^{-N}.$$

The general formula is written as follows:

$$P_{i0}(t) = \frac{1}{i!\, G(1,1,t)} \frac{\partial^i G(0,0,t)}{\partial \xi^i} = \frac{1-\kappa}{(1-\kappa)\left(1 + \frac{v}{2} \frac{e^{(\sigma-c)t} - 1}{\sigma - c}\right) + \kappa e^{(\sigma-c)t}} P_{i*}(t),$$

$$P_{i1}(t) = \frac{1}{i!\, G(1,1,t)} \frac{\partial^{i+1} G(0,0,t)}{\partial \xi^i \partial \eta} = \frac{(1-\kappa)\frac{v}{2} \frac{e^{(\sigma-c)t} - 1}{\sigma - c} + \kappa e^{(\sigma-c)t}}{(1-\kappa)\left(1 + \frac{v}{2} \frac{e^{(\sigma-c)t} - 1}{\sigma - c}\right) + \kappa e^{(\sigma-c)t}} P_{i*}(t),$$

where $i = 0, 1, \ldots N$, and the function:

$$P_{i*}(t) = \binom{N}{i} \left(\frac{u}{2N} \frac{e^{st} - 1}{s}\right)^i \left(1 + \frac{u}{2N} \frac{e^{st} - 1}{s}\right)^{-N},$$

defines the probability to pick a cell with $i$ drivers independently of resistant status, $P_{i*}(t) \triangleq P_{i0}(t) + P_{i1}(t)$, where $\binom{N}{i}$ denotes a binomial coefficient, equal $\frac{N!}{i!(N-i)!}$.

The distribution $P_{i*}(t)$ for a particular case of $N = 6$ is shown in **Appendix 1—figure 3**.

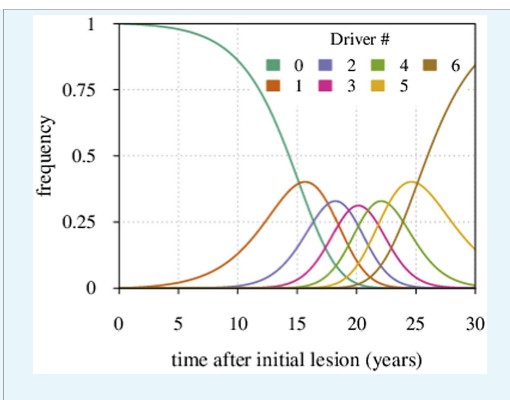

**Appendix 1—figure 3**. Maximal tumour heterogeneity in terms of driver subclones occurs at intermediate times after initial lesion.

We now derive the mean time period when a given subclone with $i$ additionally accumulated drivers dominates within a tumour.

Defining the time moments $t = t_i$ for which $P_{i-1,*}(t_i) = P_{i*}(t_i)$ ($i = 0, 1, 2,…,N$) gives:

$$t_i = \frac{1}{s}\ln\left(1 + \frac{2sNi}{u(N-i+1)}\right) \approx \frac{1}{s}\ln\frac{2sNi}{u(N-i+1)},$$

where we assume $u \ll s$.

The time period when the subclone with $i$ drivers prevails in a cell population is defined by the following expression:

$$\Delta t_i = t_{i+1} - t_i = \frac{1}{s}\left[\ln\left(1 + \frac{2sN(i+1)}{u(N-i)}\right) - \ln\left(1 + \frac{2sNi}{u(N-i+1)}\right)\right] \approx \frac{1}{s}\ln\frac{(N-i+1)(i+1)}{(N-i)i},$$

where $i \neq 0$. For $i = 0$, we have:

$$\Delta t_0 = \frac{1}{s}\ln\left(1 + \frac{2s}{u}\right) \approx \frac{1}{s}\ln\frac{2s}{u}.$$

The latter formula has been previously reported (see Equation S7 in reference **Bozic et al., 2010**).

## Varying mutation rate and initial tumour size

To understand better how the inflection points emerge around $\sigma = qs$ ($q = 1, 2, 3,…$), we first consider a much simpler case than the main text. Suppose that no additional driver mutation is acquired during tumourigenesis, $u = 0$. Any treatment regime of constant intensity that lowers the selective advantage ($\sigma < s$) only slows tumour growth. In such cases, the median detection time increases with $\sigma$ (see **Figure 6**). Assuming that resistance is not obtained, the median time to detection approaches a vertical asymptote at $\sigma = s$, whereas the tumour is always eradicated for $\sigma > s$. If $v > 0$, then the tumour relapses following the appearance of the resistance mutation, and the median detection time approaches the horizontal asymptote for $\sigma \gg s$. Thus, the final result for the median detection time will be a line with an inflection point near $\sigma = s$, consisting of two effects: higher treatment levels resulting in the control of sensitive clones with ever higher numbers of drivers (blue colouring), and the lack of control of costly resistant clones (red colouring) (**Figure 6**).

In the general case ($u > 0$, $v > 0$), the median follows from the different elements cited above, consisting of several possible branches (e.g., **Figure 2—figure supplement 1D**) or none, where

the latter case depends on the relation between the mutation rate $u$ and the initial cell number $M_0$ (e.g., for $M_0 = 10^7$ see **Figure 2—figure supplement 1B**).

**Appendix 1—figure 4** shows an example. Panel A shows a case when $u$ is varied and $M_0$ remains fixed. We identify the key parameter for the appearance of the inflection points as $M_0 u$. It defines the emergence probability of the next subclone with one additional driver. Hence, the inflection points are more apparent for smaller values of $M_0 u$ and vanish for larger values of $M_0 u$. This same tendency can be seen in **Figure 2—figure supplement 1B**, where $M_0$ is varied and $u$ is fixed. Panels B, C, and D provide more details on the interplay between $M_0 u$ and the inflection points.

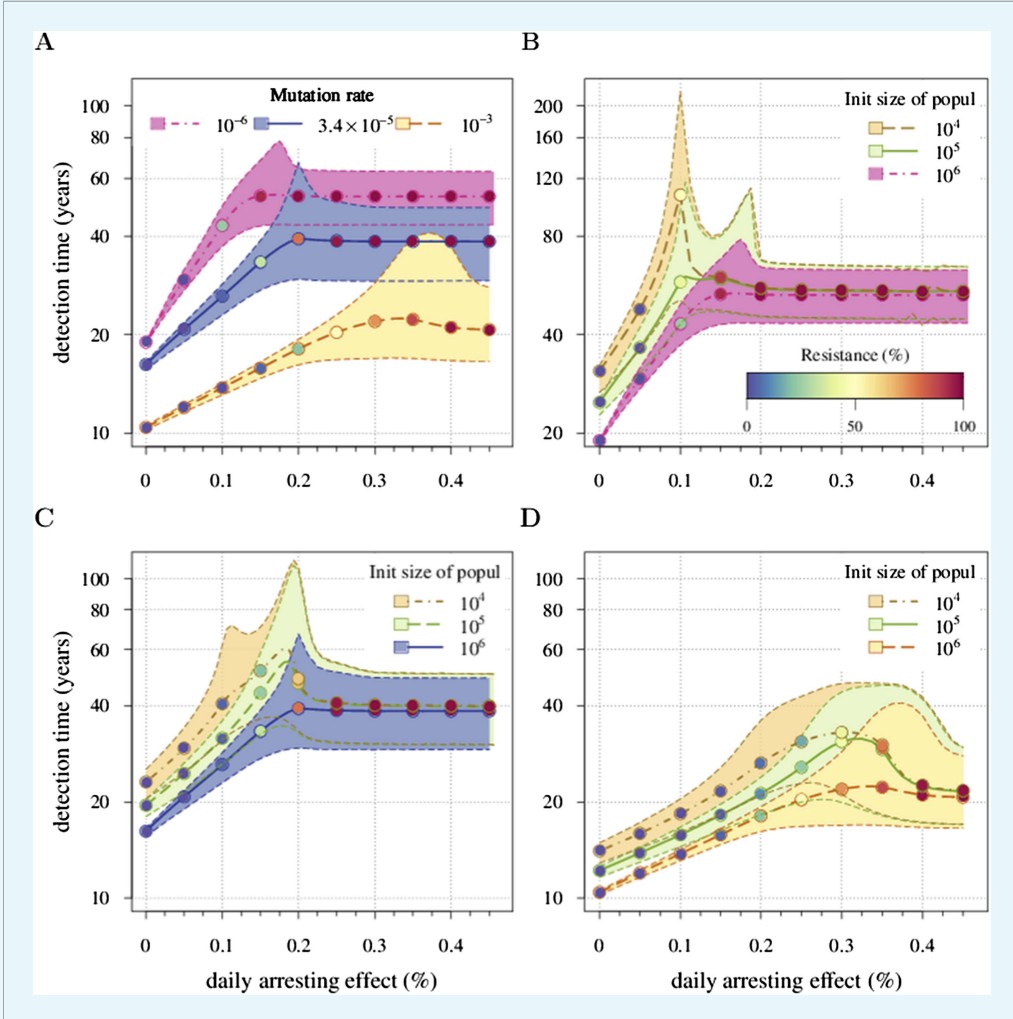

**Appendix 1—figure 4**. Sensitivity analysis of (**A**) the mutation rate to acquire drivers and (**B–D**) initial tumour size. Thick lines indicate the median and shaded areas with dashed boundaries the 90% confidence intervals of detection times. (**A**) Initial tumour size is fixed at $10^6$ cells and mutation rate is varied. (**B–D**) Initial tumour size is varied and mutation rate is fixed at (**B**) $10^{-6}$, (**C**) $3.4 \times 10^{-5}$, (**D**) $10^{-3}$. The colour code for points indicates the average level of resistance within tumours (see the inset in **B**). Parameter values as in **Table 1** except those being varied.

# A simple form of drug addiction for resistant cell-lines

To explore the possible effects of drug addiction on tumour growth, we propose a simple modification of the fitness function. Suppose the cost of resistance C is given by the relation:

$$C = c\left(1 - \frac{2\sigma}{s}\right).$$

It equals $c$ as before when no treatment is applied ($\sigma = 0$), while drug addiction increases with larger $\sigma$. The cost equals zero at $\sigma = 2s$ and becomes negative for $\sigma > 2s$, implying that further drug administration has a beneficial effect on proliferation of resistant cells. Fischer and colleagues (**Fischer et al., 2015**) argue that under drug addiction, a metronomic treatment strategy is more beneficial than a constantly applied treatment. The metronomic strategy imposes a simple rule: treatment is only applied when the number of non-resistant cells exceeds the number of resistant cells.

We compare the originally studied model, where the resistant cells exhibit no drug addiction, with a modified case of resistant cell drug addiction. In the latter, we consider two different treatment regimes: a constantly administrated drug application and metronomic therapy. As an example, **Appendix 1—figure 5** shows the occurrence drug addiction worsens treatment outcomes with 5% vs 31% eradicated tumours at a daily arresting effect of 0.25% ($\sigma = 1.0$%) for presence vs absence of drug addiction, respectively. Moreover, metronomic therapy results in a better outcome than a constantly administrated chemotherapy. For example, the former adds 6.6 additional years to the median detection time at $\sigma = 1.0$%, compared to the constant treatment with drug addiction, but loses 11.1 years, compared to the previous case of a constant treatment without drug addiction.

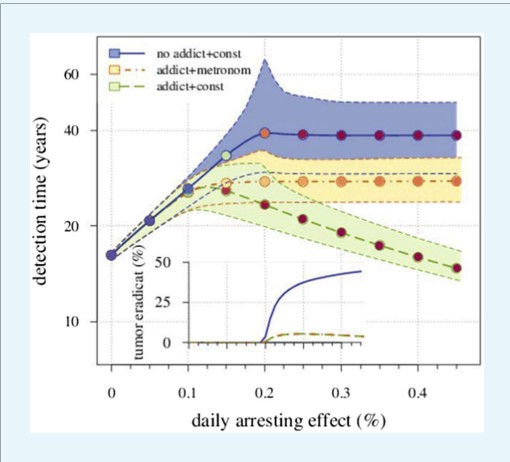

**Appendix 1—figure 5**. Comparing the case shown in blue in **Figure 2** with the case where resistant cells may become addicted to the drug. The latter is illustrated by two treatment regimes: metronomic strategy (yellow) and constantly administrated drug (green). The plot shows the median and 90% confidence intervals (shaded areas) of detection times. The inset presents the fraction of cases when the tumour goes extinct after the initial lesion of $10^6$ cells. Parameters as in **Table 1**.

