## [Decision Letter]

Thank you for sending your work entitled “Dynamics of preventive and reactive cancer control using low-impact measures” for consideration at *eLife*. Your article has been favorably evaluated by Diethard Tautz (Senior editor) and three reviewers, one of whom, Carl Bergstrom, is a member of our Board of Reviewing Editors.

The Reviewing editor and the other reviewers discussed their comments before we reached this decision, and the Reviewing editor has assembled the following comments to help you prepare a revised submission.

All three reviewers found this to be an interesting and timely paper that addresses a very important question. The work is well-conceived, attentive to detail, and well-interpreted. Nonetheless the reviewers raised a number of points which they felt should be addressed.

1) The authors use specific parameter values for their investigations. These parameters are presented in Table 1, and instead of giving biologically applicable ranges, the authors use particular values. In my opinion, it is important to investigate the range and not just a set of values. The paper would benefit greatly from some degree of sensitivity analysis to explain how results change as parameters change. For example, it is assumed that mutations are acquired with the rate 3.4x10^(^-5^). In reality, this parameter can vary over a very large range. It could be as low as 10^(^-7^) and as high as 10^(^-2^) in the presence of genomic instability. This is a change over five orders of magnitude, and can affect the results in a significant way. This should be addressed and discussed.

2) The authors assume that resistance presents a cost. At the same time, it has been suggested in the literature that some resistant mutations can actually be advantageous (e.g. in chronic lymphocytic leukemia). How will this alter the results?

3) It would be useful to discuss how the model applies or could be modified to apply to different types of therapies. Traditional therapies, more modern targeted therapies, and various sorts of combination treatments have very different properties. Often the source of resistance is not known. It is sometimes associated with specific mutations, and sometimes it has a different origin. The authors should explain how their model can be used in these different circumstances, and what modifications in the model correspond to different cases. Here we are looking less for additional modeling than for guidance on how the model could be adapted in future work.

4) Similarly, different cancers have different properties (solid tumors vs. leukemias, for example). It would be useful to discuss how future studies could extend the model to explicitly model different cancer types.

5) In the Abstract and throughout the paper, please clearly specify whether this paper is intended to address the implications of preventive/therapeutic interventions on the fate of a single aberrant clone, or on the fate of an individual patient. This distinction is important because cancer at almost any site in the body occurs in the context of additional subclinical aberrations in the population of cells. That is, most cancers arise in organs that are predisposed to cancer due to an interplay of inherited and exogenous influences (i.e., gene x environment interactions). Thus the process can be manifest not only in a single aberrant clone in one organ, but many concomitant clones within or across organs (e.g., long-term tobacco exposure places one at risk for myriad cancers in various organs) leading to synchronous and metachronous precancers and cancers.

6) The manuscript continually refers to “residual cells,” but the goal of most cancer-related surgeries is complete tumor removal. Complete resection is the foundation of cancer care wherever feasible. So it appears that this paper really relates to the fraction of cancer cases with residual or metastatic cancer. If so, that should be clarified in the title and background of the work (e.g. Dynamics of preventive and therapeutic interventions in patients with residual/metastatic cancer using low-impact measures).

---

## [Author Response]

*1) The authors use specific parameter values for their investigations. These parameters are presented in*
Table 1*, and instead of giving biologically applicable ranges, the authors use particular values. In my opinion, it is important to investigate the range and not just a set of values. The paper would benefit greatly from some degree of sensitivity analysis to explain how results change as parameters change. For example, it is assumed that mutations are acquired with the rate 3.4x10^(*^*-5*^*). In reality, this parameter can vary over a very large range. It could be as low as 10^(*^*-7*^*) and as high as 10^(*^*-2*^*) in the presence of genomic instability. This is a change over five orders of magnitude, and can affect the results in a significant way. This should be addressed and discussed*.

We used a canonical set of parameters, many based on biologically grounded estimates. We did in fact conduct sensitivity analyses for several of the most important parameters (e.g., Figure 2 and its supplements). In reply to the reviewer’s request, we have now added an additional column to Table 1, indicating plausible ranges for parameter values, based on the literature. In addition, we conducted further sensitively analyses by varying mutation rates, finding that tumors exhibit more or less deterministic growth depending on the relation between the initial number of cells and additional driver mutation rate. This is analogous to the phenomenon shown in Figure 2—figure supplement 1, whereby the higher the mutation rate, the less apparent are stochastic effects. The corresponding analysis is presented in Section B of Appendix 1 (see also Figure 5). This is referenced by a paragraph in the Results, in the Preventive measures section.

*2) The authors assume that resistance presents a cost. At the same time, it has been suggested in the literature that some resistant mutations can actually be advantageous (e.g. in chronic lymphocytic leukemia). How will this alter the results*?

This is a good point, and we agree that no cost, or more extreme situations such as drug addiction in resistant cell-lines (e.g., in certain melanomas (Das Thakur et al., 2014) or in leukemias (Michor et al., 2005), is important to consider for tumor management. Establishing optimized schedules in such situations is an intriguing problem that we plan to study in more detail in the future. In reply to the reviewers’ comment, we considered how drug addiction might alter our results. See Section C of Appendix 1. The corresponding paragraph in the main text is added to the end of the Results, Preventive measures section,. We also added two references (Das Thakur et al., 2014 and Fischer, Vásquez-García and Mustonen, 2015).

*3) It would be useful to discuss how the model applies or could be modified to apply to different types of therapies. Traditional therapies, more modern targeted therapies, and various sorts of combination treatments have very different properties. Often the source of resistance is not known. It is sometimes associated with specific mutations, and sometimes it has a different origin. The authors should explain how their model can be used in these different circumstances, and what modifications in the model correspond to different cases. Here we are looking less for additional modeling than for guidance on how the model could be adapted in future work*.

Our current discussion does discuss cytotoxic vs. cytostatic effects (please see the subsection headed “Preventive interventions”). Although we agree that discussing model extensions to other types of therapy would be useful, it is unclear without actually modifying the model how these would alter the results, and as such we simply mention the importance of future study to consider these alternatives (in our Conclusion, we state: “Moreover, future studies should also investigate alternatives to the traditional post-diagnostic therapeutic scenarios considered here [e.g., molecularly targeted therapies (Yap, 2015)]”). We do however think it is important to stress the mutational basis for resistance, and as such have added text in the Discussion.

*4) Similarly, different cancers have different properties (solid tumors vs. leukemias, for example). It would be useful to discuss how future studies could extend the model to explicitly model different cancer types*.

The second paragraph of the Conclusions now provides a series of possible modifications that can be made to our model to consider different cancer types and levels of resolution. Necessary references were also added.

*5) In the Abstract and throughout the paper, please clearly specify whether this paper is intended to address the implications of preventive/therapeutic interventions on the fate of a single aberrant clone, or on the fate of an individual patient. This distinction is important because cancer at almost any site in the body occurs in the context of additional subclinical aberrations in the population of cells. That is, most cancers arise in organs that are predisposed to cancer due to an interplay of inherited and exogenous influences (i.e., gene x environment interactions). Thus the process can be manifest not only in a single aberrant clone in one organ, but many concomitant clones within or across organs (e.g., long-term tobacco exposure places one at risk for myriad cancers in various organs) leading to synchronous and metachronous precancers and cancers*.

http://tiny.cc/AkhmHoch15SVideos45We now clarify that we are examining the simple situation of the fate of an individual patient confronted with single cancer. We have added text to the Abstract and Introduction clarifying this, and at the end of the Discussion stressed that future models should consider multiple independent or correlated events (same or different organs).

*6) The manuscript continually refers to* “*residual cells,*” *but the goal of most cancer-related surgeries is complete tumor removal. Complete resection is the foundation of cancer care wherever feasible. So it appears that this paper really relates to the fraction of cancer cases with residual or metastatic cancer. If so, that should be clarified in the title and background of the work (e.g. Dynamics of preventive and therapeutic interventions in patients with residual/metastatic cancer using low-impact measures)*.

Prevention in our model acts on cancers before they are detected. What we now call “post-diagnostic interventions” (“reactive measures” in the original manuscript) involve resection followed by treatment. We assume that either resection itself is not perfect, and/or neighboring micro-metastases exist, and/or distant metastases are present, but undetected or non-removable. We show how the population number and (epi)genetic composition (in terms of drivers and resistance) of residual cells can have a major impact on both preventive (but initially failed) and post-diagnostic interventions (Figure 6 vs. Figure 6). We now clarify this in the manuscript (please see the Introduction and the subsections entitled “Post-diagnostic interventions” and “Prevention vs. post-diagnostic intervention”).